# Simultaneous profiling of histone modifications and DNA methylation via nanopore sequencing

Xue Yue[1,7], Zhiyuan Xie[1,7], Moran Li[1], Kai Wang[1], Xiaojing Li[1], Xiaoqing Zhang [2,3,4], Jian Yan [5,6] & Yimeng Yin [1,2] ✉

The interplay between histone modifications and DNA methylation drives the establishment and maintenance of the cellular epigenomic landscape, but it remains challenging to investigate the complex relationship between these epigenetic marks across the genome. Here we describe a nanopore-sequencing-based-method, nanoHiMe-seq, for interrogating the genome-wide localization of histone modifications and DNA methylation from single DNA molecules. nanoHiMe-seq leverages a nonspecific methyltransferase to exogenously label adenine bases proximal to antibody-targeted modified nucleosomes in situ. The labelled adenines and the endogenous methylated CpG sites are simultaneously detected on individual nanopore reads using a hidden Markov model, which is implemented in the nanoHiMe software package. We demonstrate the utility, robustness and sensitivity of nanoHiMe-seq by jointly profiling DNA methylation and histone modifications at low coverage depths, concurrently determining phased patterns of DNA methylation and histone modifications, and probing the intrinsic connectivity between these epigenetic marks across the genome.

Histone modifications and DNA methylation are fundamental epigenetic marks that contribute to distinct gene expression programs and biological functions[1]. Instead of acting independently of each other, these different modifications are closely connected and the crosstalk between them plays a crucial role in the establishment of chromatin diversity, with complex modification patterns resulting in distinct functional outcomes[2]. In the past decades, various approaches, such as chromatin immunoprecipitation followed by high-throughput sequencing (ChIP-seq), cleavage under targets and tagmentation (CUT&Tag) and whole-genome bisulfite sequencing (WGBS), have been developed to map epigenetic marks across the genome[3–5]. Although they are powerful techniques, individual methods are only capable of profiling either histone modifications or DNA methylation, and the relationship between these two types of epigenetic marks uncovered by such techniques is complicated by many factors, such as cell population heterogeneity and allele-specific chromatin marking. Recently, several promising techniques have been developed to jointly analyze chromatin features in the same DNA molecules in a single assay. These techniques include sequential ChIP-bisulfite-sequencing (ChIP-BS-seq), CUT&Tag coupled with tagmentation-based bisulfite sequencing (CUT&Tag-BS), and nucleosome occupancy and methylome-sequencing in single cells (scNOMe-seq)[6–8]. However, these methods all rely on bisulfite conversion to distinguish methylated from unmethylated cytosines, and hence, they have the

[1]Translational Research Institute of Brain and Brain-Like Intelligence, Shanghai Fourth People's Hospital, School of Medicine, Tongji University, Shanghai 200434, China. [2]Clinical Center for Brain and Spinal Cord Research, Tongji University, Shanghai 200092, China. [3]Translational Medical Center for Stem Cell Therapy, Shanghai East Hospital, School of Medicine, Tongji University, Shanghai 200120, China. [4]Key Laboratory of Spine and Spinal Cord Injury Repair and Regeneration of Ministry of Education, Orthopaedic Department of Tongji Hospital, Shanghai 200065, China. [5]School of Medicine, Northwest University, Xi'an 710069, China. [6]Tung Biomedical Sciences Centre, Department of Biomedical Sciences, City University of Hong Kong, Kowloon, Hong Kong, SAR, China. [7]These authors contributed equally: Xue Yue, Zhiyuan Xie. ✉e-mail: yy461@tongji.edu.cn

disadvantages of DNA damages, complexity reduction and biases introduced by the bisulfite treatment. The resulting libraries from these techniques do not adequately cover the genome or the targeted regions, especially the sites weakly bound by target proteins and the sites with higher CpG and/or GpC frequencies[9]. The techniques that use enzymatic reactions to distinguish methylated cytosines (5mCs), such as enzymatic methyl-seq (EM-seq), overcome the limitations of bisulfite treatment[9]. However, these techniques are hindered by the difficulties in preparing hyperactive enzyme(s) and the methylation measurements are often confounded by the incomplete conversion of methylated or unmethylated cytosines. In addition, we are not aware of any technique that combines ChIP-seq or CUT&Tag with enzymatic methods to simultaneously explore DNA methylation and histone modifications in a single assay.

Recently, Oxford Nanopore Technologies (ONT) have been widely used to detect different forms of DNA methylation, such as 5-mC at CpG sites and N6-methyldeoxyadenosines (6mAs) in the sequence of GATC. These forms of methylation can be detected in long sequencing reads by carefully analyzing the electrical current signals measured by nanopore-based sequencing devices[10,11]. Based on this advantage, various techniques have recently been developed to explore the chromatin features by exogenously introducing methyl groups to the DNA bases of target regions. For instance, nanoNOMe was developed to jointly evaluate CpG methylation and chromatin accessibility after exogenously labeling GpC sites at open chromatin regions, and DiMeLo-seq was developed to jointly analyze CpG methylation and the binding of proteins of interest by leveraging deoxyadenosine methyl-transferase to mark adenines proximal to target proteins[12,13]. However, the utility of nanoNOMe to investigate chromatin features is limited by factors such as the sporadic occurrence and linear clustering of GpC dinucleotides and the endogenous cytosine methylation[14]. Due to its lack of endogenous methylation and its much higher occurrence frequency, at almost one in every two DNA base pairs, adenine is an ideal candidate for marking regions of interest across the human genome[14]. Unfortunately, only the ONT-provided computational tools Megalodon and Guppy are currently available to identify 6mAs in all possible

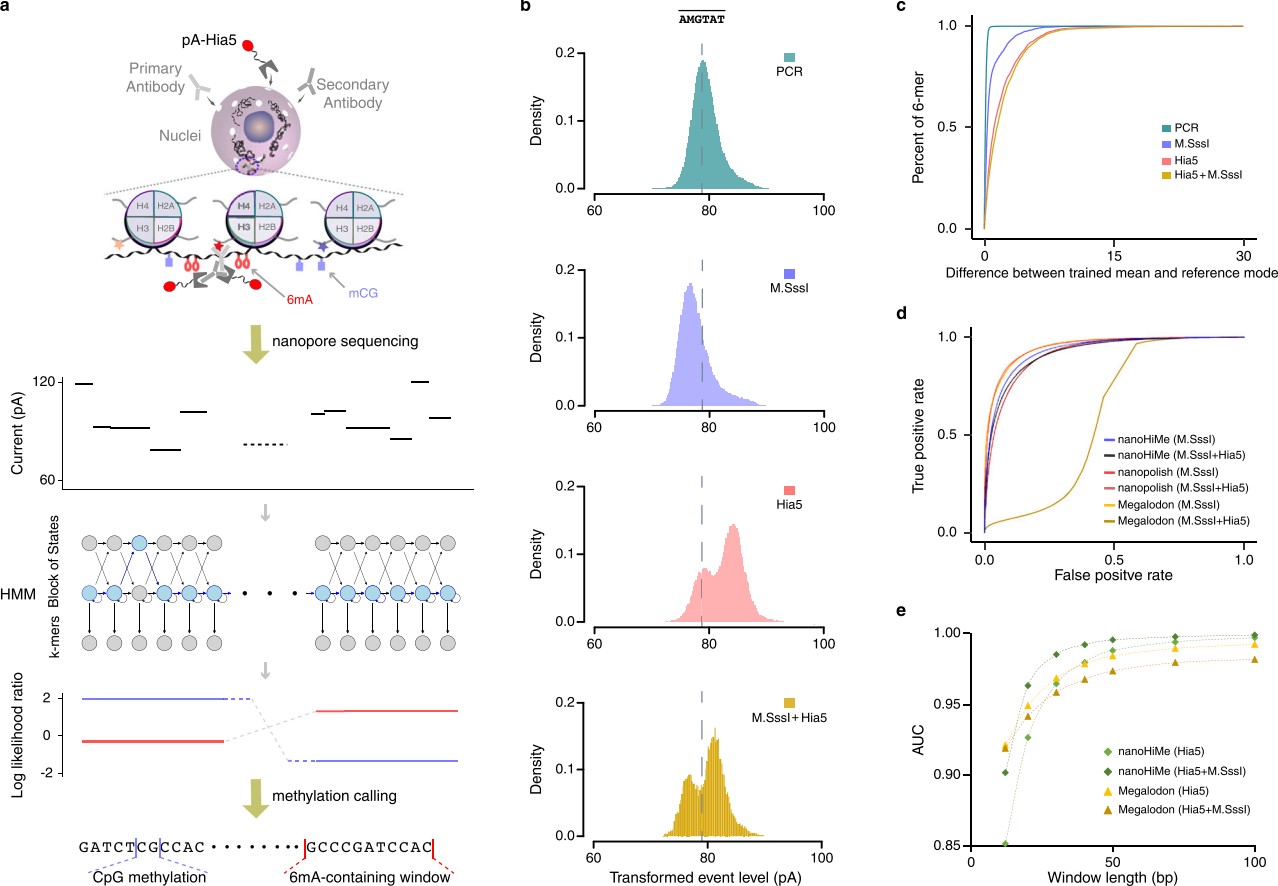

**Fig. 1 | Schematic and the performance of nanoHiMe-seq. a** The steps in nanoHiMe-seq. A primary antibody (light gray) binds to modified nucleosomes, a secondary antibody (medium gray) binds to the primary antibody, and both antibodies recruit the pA-Hia5 fusion protein (dark gray and red) to the targeted sites. The tethered pA-Hia5 methylates the adenines (6mAs; red oval) nearby. After sequencing, basecalling and alignment to the reference genome in base- and event-space, the likelihood of a sequence having or lacking modified base(s) is calculated for individual reads using a hidden Markov model. The ratio of the likelihood is used to identify methylated CpG sites (blue rectangle) and/or 6mA-containing sites. **b** Differences in the event distribution for a 6-mer with or without modified base(s). Nanopore sequencing data used to plot the event distribution of $\overline{AMGTAT}$ was from PCR amplicons without treatment (cyan), treated by M.SssI (blue), Hia5 (red) or both M.SssI and Hia5 (yellow). M indicates 5mC and $\overline{AMGTAT}$ represents a collection of *k*-mers derived from AMGTAT, but with 6mA at the first position, fifth position or both positions. **c** An overview of the differences between the reference models and the means of trained Gaussian using data from PCR amplicons without treatment (cyan), treated with M.SssI (blue), Hia5 (red), or both M.SssI and Hia5 (yellow). For 6-mers with more than one set of parameters, the mean with the most significant shift from the reference model was selected for the calculation. **d** Benchmarking of nanoHiMe, nanopolish and Megalodon for mCpG detection. A receiver operating characteristic (ROC) curve was used to assess the performance of different computational tools at calling CpG methylation on nanopore reads from PCR amplicons without treatment, treated by M.SssI, or both M.SssI and Hia5. **e** Assessment of the performance of nanoHiMe and Megalodon at identifying 6mA-containing sites of different lengths. The AUC was calculated based on the calls from nanoHiMe or Megalodon for each group of sites, and plotted as a function of the length of the sites. Source data are available in the Source Data file.

contexts and 5mCs in CpG context from individual nanopore reads. Moreover, the performance of these tools at jointly calling 6mA and 5mC from a single DNA molecule hasn't been systematically evaluated.

Here, we describe a method - nanoHiMe-seq to mark adenines around modified nucleosomes of interest and provide a computational tool to identify the modified adenines and endogenous 5mCs in individual nanopore sequencing reads. We systematically evaluate the performance of our computational tool and ONT-provided tool, and demonstrate the utility, robustness and sensitivity of nanoHiMe-seq by jointly profiling CpG methylation and representative histone modifications of heterochromatin and euchromatin. Taking advantage of the long reads generated by nanopore sequencing, we further use nanoHiMe-seq to identify allele-specific epigenetic states across the genome and to probe the intrinsic connectivity between epigenetic marks along multikilobase segments of the genome. We anticipate that nanoHiMe-seq will be widely applied to investigate the functional coordination of epigenetic marks in various biological contexts and to explore chromatin features in complex genomic regions.

## Results

### Overview of nanoHiMe-seq

Nanopore sequencing has the advantage of distinguishing various DNA modifications through careful analysis of electric current events. Based on this advantage, we developed a **nano**pore-sequencing-based **Hi**stone-modification and **Me**thylome joint-profiling method, named nanoHiMe-seq. In nanoHiMe-seq, the nuclei were permeabilized and the modified nucleosomes of interest in the nuclei were first bound in situ with a specific primary antibody. The primary antibody was then bound by a secondary antibody, and both antibodies subsequently tethered a protein A−N6-adenine methyltransferase (Hia5) fusion protein (pA-Hia5) to the modified nucleosomes. After the unbound components were washed away, pA-Hia5 was activated by the addition of S-adenosylmethionine (SAM) and the adenines proximal to the target sites were methylated (Fig. 1a, top). Genomic DNA was extracted from the nuclei and subjected to nanopore sequencing (Fig. 1a, middle). After basecalling and the alignment of sequencing reads to the reference genome, the electric current events from each read were re-analyzed with a hidden Markov model (HMM), which was implemented in the nanoHiMe software package, to determine whether a site contained 6mAs and to call CpG(s) at the site as methylated or unmethylated (Fig. 1a, bottom).

### nanoHiMe-seq performance

To ensure that nanoHiMe-seq was feasible, we first produced the pA-Hia5 fusion protein and tested its enzymatic activity (Supplementary Fig. 1a, b). We found the treatment of DNA templates with increasing amounts of pA-Hia5 resulted in monotonic increases in adenine methylation (Supplementary Fig. 1b). Moreover, the methyltransferase activity was not hindered by the presence of the methyl group in CpG dinucleotides adjacent to adenine bases (Supplementary Fig. 1c). Thus, pA-Hia5 is a promising candidate for probing modified nucleosomes surrounded by methylated or unmethylated DNA of all possible sequences.

To determine whether a site on a nanopore read contained 6mA(s), and to call CpG(s) on the read as methylated or unmethylated with an HMM, we needed to learn the parameters of emission distribution for individual $k$-mers with 5-mC at CpG sites and 6mA in all possible contexts (see Methods). Therefore, we generated four datasets from PCR-amplified genomic DNA of *Escherichia coli* K12 MG1655 (Supplementary Data 2) that was either untreated (PCR$^+$ M.SssI$^-$ Hia5$^-$) or treated with methyltransferase enzye M.SssI (PCR$^+$ M.SssI$^+$ Hia5$^-$), Hia5 (PCR$^+$ M.SssI$^-$ Hia5$^+$) or both (PCR$^+$ M.SssI$^+$ Hia5$^+$), resulting in samples free of methylation, with near-complete CpG methylation (96.2%) and/or partial adenine methylation, respectively. It is worth noting that we learned the parameters from DNA templates with

partial adenine methylation, which enabled us to obtain the parameters of $k$-mers with 6mAs in all possible contexts. However, partial adenine methylation made it impossible to precisely assign the learned parameters to $k$-mers that were derived from a four-letter $k$-mer, but contained 6mA(s) at different positions, such as TZCACG, TACZCG and TZCZCG, where Z denotes 6mA. To overcome this limitation, we grouped such $k$-mers as a new $k$-mer, $\bar{k}$, and assigned the parameters from these $k$-mers to $\bar{k}$ (Supplementary Fig. 2). As a result, nanoHiMe gained the ability to accurately identify the sites with fully methylated adenines, a mixture of methylation patterns, or no methylation as a 6mA-containing site or a non-6mA site, but lost the ability to predict adenine methylation at base resolution.

To assess the performance of our training step, we first compared our learned parameters from a DNA sample lacking methylation or treated with M.SssI to the corresponding parameters of individual $k$-mers from nanopolsih, a widely used computational tool that applies an HMM to call 5mC in CpG and GpC contexts[15]. It was found that the parameters learned from nanoHiMe were highly consistent with those provided by nanopolish (Supplementary Fig. 3a). Additionally, we trained the parameters of emission distribution for individual $k$-mers using signalAlign[16] in DNA samples treated with M.SssI, Hia5, or both, which were previously used by nanoHiMe for parameters learning. We found that the parameters learned using nanoHiMe also correlated well with those learned from signalAlign for each DNA sample (Supplementary Fig. 3b). When analyzing the obtained parameters, we found the shifts in the electrical signals for a considerable number of $k$-mers after treatment with M.SssI, Hia5, or both (Fig. 1b, c). The shifts were prone to be more significant when adenine methylation occurred in adjacent bases and in the middle of the $k$-mers (Supplementary Fig. 3c–e). This differs from CpG methylation, in which the most significant shifts were observed when the methylation occurred at the fifth position of a 6-mer[10,13].

We then applied our trained models to calculate the likelihood that the underlying nucleotides sequence was a methylated or an unmethylated version of a genome substring when a sequence of current events was observed. Similar to previous studies[10,13], we used the log likelihood ratio (LLR) to make a methylation call for each site on individual nanopore reads. To evaluate the accuracy of the methylation calls, we randomly sampled 100,000 singleton sites, i.e., regions that contained only a single CpG, from each of unmethylated and methylated *E. coli* DNA samples. We considered the sites from PCR$^+$ M.SssI$^-$ Hia5$^-$ and PCR$^+$ M.SssI$^-$ Hia5$^+$ datasets as unmethylated, and those from PCR$^+$ M.SssI$^+$ Hia5$^-$ and PCR$^+$ M.SssI$^+$ Hia5$^+$ datasets as methylated. We then calculated the number of these sampled sites that were correctly identified as methylated or unmethylated when the LLR was greater or smaller than a specific threshold and plotted receiver operating characteristic (ROC) curves across a range of LLR thresholds (Fig. 1d). To compare the performance of different computational tools, we also calculated the number of sampled sites that were correctly classified as methylated or unmethylated by Megalodon and nanopolish, and plotted ROC curves for a range of probability thresholds (Megalodon) or LLR thresholds (nanopolish). To quantitatively compare the performance of nanoHiMe, nanopolish and Megalodon, we calculated the area under their ROC curves (AUCs) and found that Megalodon performed slightly better than nanopolish and nanoHiMe in predicting CpG methylation of the sampled sites that did not contain 6mA(s) (Fig. 1d; Megalodon AUC: 0.953, nanopolish AUC: 0.951 and nanoHiMe AUC: 0.934). However, when calling CpG methylation of the sites that also contained 6mA(s), nanoHiMe and nanopolish performed markedly better than Megalodon, with nanoHiMe exhibiting the best performance (Fig. 1d; nanoHiMe AUC: 0.922, nanopolish AUC: 0.919 and Megalodon AUC: 0.603).

We additionally compared the performance of nanoHiMe and Megalodon in identifying 6mA-containing sites. We first sampled seven groups of sites, with lengths ranging from 12 bp to 100 bp, from each

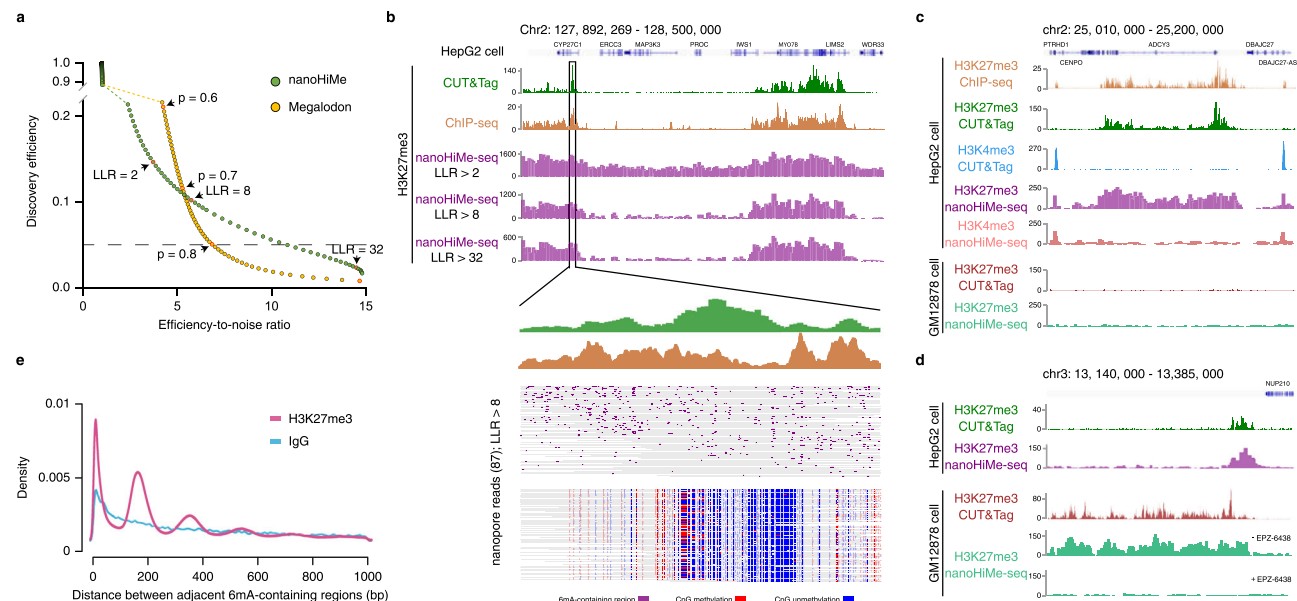

**Fig. 2 | nanoHiMe-seq for histone modification profiling. a** Comparison of the performance of nanoHiMe and Megalodon at identifying 6mA-containing sites from nanoHiMe-seq data. **b** View of 6mA signals in H3K27me3 nanoHiMe-seq data across a 500-kb segment of the human genome. The numbers of 6mA-containing sites identified by nanoHiMe using different log likelihood ratio (LLR) cut-offs (2, 8, and 32) were counted and plotted in 2500-bp windows. H3K27me3 CUT&Tag and ChIP-seq signals in the region are also shown on the top of the panel. A magnified view is shown for CUT&Tag and ChIP-seq peaks, and the predicted 6mA-containing sites (purple), methylated CpG sites (red) and unmethylated CpG sites (blue) in individual nanopore reads from H3K27me3 nonaoHiMe-seq. **c, d** View of 6mA signals from H3K27me3 and H3K4me3 nanoHiMe-seq in HepG2 and GM12878 cells. Similar to (b), the numbers of 6mA-containing sites detected in each experiment using an LLR cut-off of 32 were counted and plotted in 1250-bp windows. H3K27me3 and H3K4me3 signals from CUT&Tag and ChIP-seq are also shown. **e** Distribution of the distances between two adjacent 6mA-containing sites in individual nanopore reads from either H3K27me3 nanoHiMe-seq (purple) or nanoHiMe-seq using non-specific IgG (blue). The nanopore reads overlapping the top 50% of H3K27me3 peaks were selected for the analysis. Source data are available in the Source Data file.

of *E. coli* DNA samples with or without 6mA. We next calculated the area under the ROC curves, which were plotted using the calls from nanoHiMe or Megalodon for each group of sites. We found that the performance of both nanoHiMe and Megalodon was highly correlated with the length of the sites assessed, with better performance in predicting adenine methylation for longer sites (Fig. 1e). When evaluating sites that did not harbor methylated CpG(s) and were no shorter than 40 bp, nanoHiMe performed slightly better than Megalodon (Fig. 1e, nanoHiMe AUC 40 bp: 0.98, 50 bp: 0.988, 72 bp: 0.994, 100 bp: 0.997; Megalodon AUC 40 bp: 0.979, 50 bp: 0.984, 72 bp: 0.99, 100 bp: 0.992). When the evaluated sites also contained mCpG(s), nanoHiMe substantially outperformed Megalodon (Fig. 1e, nanoHiMe AUC 40 bp: 0.992, 50 bp: 0.995, 72 bp: 0.997, 100 bp: 0.999; Megalodon AUC 40 bp: 0.967, 50 bp: 0.974, 72 bp: 0.98, 100 bp: 0.982). We additionally compared the performance of nanoHiMe at predicting methylations by considering or not considering the co-influence of 6mA and 5mC, and found that it performed slightly better at identifying 6mA- and mCpG-containing sites from DNA with two types of methylations by considering the co-influence (6mA at 50-bp sites AUC: 0.995 vs 0.9799; mCG AUC: 0.9222 vs 0.9221).

**nanoHiMe-seq precisely maps histone modifications in both compacted and open chromatin**

To test whether nanoHiMe-seq can be applied to profile histone modifications in compacted chromatin and to compare the performance of nanoHiMe and Magalodon in predicting 6mA-containing sites in realistic data, we performed nanoHiMe-seq in HepG2 cells using non-specific IgG and an antibody against H3K27me3 (Supplementary Data 2), an abundant histone modification that is catalyzed by Polycomb Repressive Complex 2 (PRC2) and is associated with transcriptional repression[17,18]. To explore the 6mA-containing sites, we binned 50 bp on each nanopore read as a window and determined

whether the windows contained 6mA(s). We then quantified the proportion of the 50-bp windows that were called as 6mA-containing sites across all reads mapping to the H3K27me3 peak regions. We defined the proportion of 6mA-containing sites obtained from experiments using the anti-H3K27me3 antibody as the discovery efficiency and the proportion obtained from non-specific IgG experiments as noise. To evaluate the performance of nanoHiMe and Megalodon, we plotted the discovery efficiency as a function of the ratio of discovery efficiency to noise (as a proxy for the signal-to-noise ratio) across a range of LLR (nanoHiMe, −100–37) or probabilities (Megalodon, 0–0.9) thresholds. We found that nanoHiMe was more robust than Megalodon at identifying high-confidence 6mA-containing regions (Fig. 2a). For instance, at a discovery efficiency of 0.05, the efficiency-to-noise ratio of nanoHiMe was 10.85, which was >1.5 times higher than the efficiency-to-noise of Megalodon (6.85; Fig. 2a, dashed line).

To assess the performance of nanoHiMe in profiling histone modifications, we compared 6mA signals detected by nanoHiMe with profiles generated by CUT&Tag and ChIP-seq for H3K27me3 and H3K4me3 in HepG2 and/or GM12878 cells (Supplementray Figs. 4a, b and 5a). The display of 6mA calls from nanopore reads mapped to the human reference genome showed a clear pattern of chromatin domains marked by H3K27me3 or H3K4me3, and this pattern resembled the profiles determined by CUT&Tag and ChIP-seq in HepG2 or GM12878 cells (Fig. 2b–d; Supplementary Fig. 4d). As expected, increasing the LLR threshold from 2 to 32 resulted in significant noise reduction and thus higher signal-to-noise ratio (Fig. 2b). When examining the 6mA signals from H3K27me3 and H3K4me3 nanoHiMe-seq experiments in HepG2 cells, we observed distinct patterns, as revealed by CUT&Tag and ChIP-seq (Fig. 2c). Moreover, the distinct patterns of 6mA signals were also observed when examining H3K27me3-marked regions from the two cell lines (Fig. 2c, d). In contrast, H3k27me3 nanoHiMe-seq from GM12878 cells treated with the EZH2 inhibitor

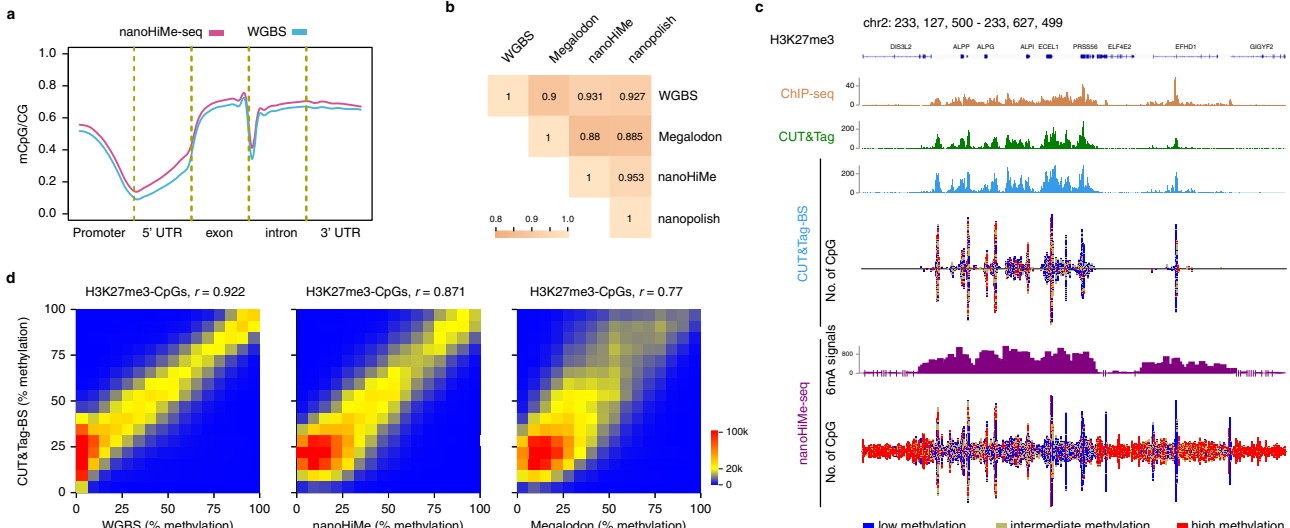

**Fig. 3 | nanoHiMe-seq for CpG methylation measurements. a** CpG methylation density revealed by nanoHiMe-seq (purple) and whole-genome bisulfite sequencing (WGBS; blue) throughout distinct gene-associated regions. **b** The performance of nanoHiMe, nanopolish, and Megalodon at calling CpG methylation from H3K27me3 nanoHiMe-seq data. The methylation levels of CpGs across the genome were calculated based on calls from nanoHiMe, nanopolish, or Megalodon from individual nanopore reads. The measurements from each of the three tools and from WGBS were compared, and the hierarchically clustered correlation matrix is shown. **c** View of H3K27me3 signals and CpG methylation status detected by nanoHiMe-seq and CUT&Tag-BS across a 500-kb segment of the human genome. The number of 6mA-containing sites identified by nanoHiMe-seq and the number of CpG sites covered by ≥5 nanoHiMe-seq and CUT&Tag-BS reads were counted

and plotted in 1250-bp windows. The CpG sites were categorized into three groups: high methylation level (red, methylation ratio ≥ 70%), intermediate methylation level (yellow, 30% ≤ methylation ratio < 70%) or low methylation level (blue, methylation ratio < 30%). The top of the panel shows H3K27me3 signals from CUT&Tag and ChIP-seq. **d** Comparison of CpG methylation levels determined by CUT&Tag-BS to those measured by WGBS, nanoHiMe or Megalodon at H3K27me3-marked regions. The $x$-axis is the methylation percentage of individual CpGs calculated from WGBS (left), or calculated based on the calls from nanoHiMe (middle) or Megalodon (right). The $y$-axis denotes the methylation percentage measured by CUT&Tag-BS. The CpG sites covered by ≥20 reads from both assays were selected for the analysis. Source data are available in the Source Data file.

EPZ6438[19], in which H3K27 methylation was severely depleted, produced very sparse 6mA signals, with a 6mA-containing site being detected in approximately every 16,400 bp (Fig. 2d, bottom; Supplementary Fig. 4e, f).

As N⁶-adenine methyltransferases (6mA-MTases) show high selectivity for accessible DNA, but not for the nucleosome-wrapped DNA[20], the targeted nucleosomes should be located in the regions between two adjacent 6mA-containing sites on individual nanopore reads. Hence, we investigated the length of these regions and found the following two categories of segments that were evident in H3K27me3 peak regions: (1) the regions with the lengths representing one, two, or multiple nucleosomes and (2) more numerous shorter regions with the lengths <100 bp, paralleling the distribution of internucleosomal linker regions (Fig. 2e). This observation highlights the high resolution of nanoHiMe-seq for mapping target nucleosomes.

In contrast to H3K27me3, H3K4me3 marks active chromatin sites and is a key epigenetic mark for transcription initiation. Therefore, we investigated the distribution of 6mA signals from H3K4me3 nanoHiMe-seq around the transcription starting site (TSS). By averaging the signal across active TSSs, we observed an enrichment of 6mA signals around TSSs, which resembled profiles generated by CUT&Tag and ChIP-seq. However, unlike ChIP-seq signals, which peaked at the flanking regions of the active TSSs, the signals from nanoHiMe-seq were highly enriched at canonical nucleosome-free promoter regions overlapping the TSSs[21] (Supplementary Fig. 5a, b; Supplementary Data 2). These differences are likely explained by the fact of that the DNA fragments enriched by ChIP-seq were from H3K4me3-marked nucleosomes, whereas the 6mA-containing sites from nanoHiMe-seq were methylase-accessible linker sequences flanking modified nucleosomes (Supplementary Fig. 5c).

## nanoHiMe-seq simultaneously measures DNA methylation inside and outside of target regions

To assess the performance of nanoHiMe in calling CpG methylation on nanopore reads obtained from nanoHiMe-seq experiments, we explored CpG methylation patterns across the genome using our computational tool. Like nanopolish[10], LLR cut-offs of ±1.5 were applied to call every CpG-containing site in each nanopore read as methylated (LLR ≥ 1.5) or unmethylated (LLR ≤ −1.5) using nanoHiMe (Supplementary Data 3). We calculated the density of mCpG sites based on the calls across the genome and observed the previously documented correlation between the mCpG density and the distance from the TSSs. CpG sites tended to be unmethylated near the TSSs, with the methylation level increasing significantly towards the gene body (Fig. 3a). To compare the performance of nanoHiMe with other tools, we also made methylation calls for each CpG-containing site from nanoHiMe-seq reads using Megalodon and nanopolish. For Megalodon, we used different probability cut-offs (from 0.6 to 0.9) to make methylation calls, and subsequently calculated the methylation percentages based on the calls for individual CpG sites across the genome. By comparing with the methylation levels determined by WGBS, we found that Megalodon gave the highest consistency at a probability cut-off of 0.6 (Supplementary Fig. 6a, Pearson's $r = 0.9$), and that the methylation levels predicted by nanoHiMe correlated best with WGBS measurements (Fig. 3b; Pearson's $r = 0.931$). When comparing the methylation percentages calculated by the three nanopore-based methods for each CpG site, we found that the measurements from nanoHiMe correlated better than those from Megalodon with the methylation levels predicted by nanopolish (Fig. 3b; correlation between nanoHiMe and nanopolish: Pearson's $r = 0.953$; correlation between Megalodon and nanopolish: Pearson's $r = 0.885$).

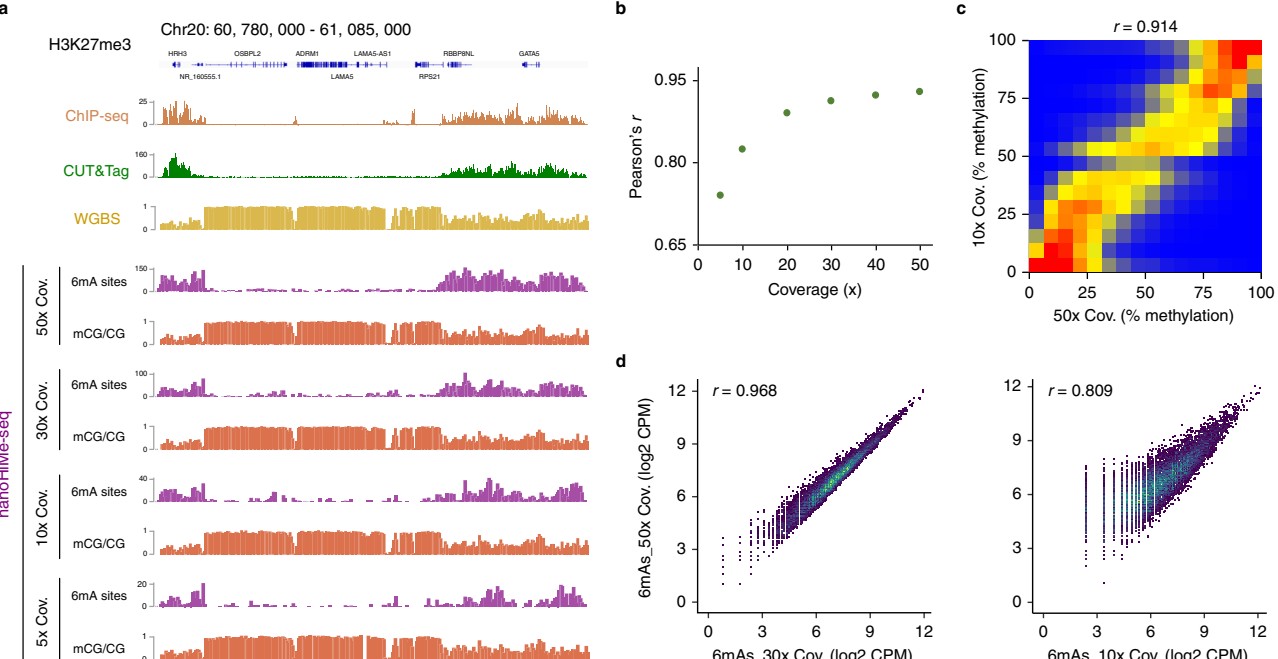

**Fig. 4 | nanoHiMe-seq requires fewer sequencing reads. a** View of 6mA signals and CpG methylation density for 50×, 30×, 10× and 5× coverage depths. The 6mA-containing sites identified using an LLR cut-off of 32 were counted in 1000-bp bins. The top of the panel shows H3K27me3 signals from CUT&Tag and ChIP-seq, as well as CpG methylation density measured by WGBS. **b** Pearson's correlation coefficients for the comparison between CpG methylation levels measured by nanoHiMe-seq with various coverage depths and measurements from WGBS. **c** Comparison of CpG methylation levels measured by nanoHiMe-seq from 50× and 10× coverage depths. **d** Density scatterplots showing the correlation between 6mA signals for 50× and 30× (left) or 10× (right) coverage depths. Each dot represents an individual peak, and the numbers of 6mA-containing sites in individual peaks were normalized as counts per million reads (CPM). Pearson's correlation coefficient is shown at the top of each plot. Source data are available in the Source Data file.

To test the power of our computational tool to simultaneously call two types of methylations from a single DNA molecule, we compared H3K27me3 profiles and methylation levels of individual CpGs at H3K27me3-marked regions, as determined by nanoHiMe-seq and CUT&Tag-BS[7]. We first assessed the quality of our CUT&Tag-BS experiments, and found that almost all peaks detected by CUT&Tag-BS overlapped with those identified in CUT&Tag experiments and that the peak signals also exhibited a high level of correlation (Fig. 3c; Supplementary Fig. 6b, d). However, we noticed that some of the weak peaks detected by CUT&Tag were not detected by CUT&Tag-BS, probably due to the DNA damages introduced by the bisulfite treatment in CUT&Tag-BS experiments (Supplementary Fig. 6b, c). When comparing the patterns of 6mA signals from nanoHiMe-seq with the profiles obtained from CUT&Tag-BS, we found that the enrichment of 6mA signals was detected in both the regions of peaks identified by CUT&Tag-BS and the regions of weak peaks identified only by CUT&Tag (Fig. 3c; Supplementary Fig. 6e, f). We next compared the methylation levels of CpG sites measured using the data from CUT&Tag-BS, nanoHiMe-seq and WGBS. Unlike the previous findings that CpG methylation levels at enhancers measured by CUT&Tag-BS were not well aligned with those obtained from WGBS in mouse embryonic stem cells[7], we found that CpG methylation levels measured by CUT&Tag-BS and WGBS were highly consistent at H3K27me3-marked regions in HepG2 cells (Fig. 3d, left panel, Pearson's $r = 0.922$). Moreover, we found that CpG methylation levels measured by nanoHiMe were also well aligned with those measured by CUT&Tag-BS (Fig. 3d, middle panel, Pearson's $r = 0.871$). We further calculated the methylation percentages for individual CpG sites based on methylation calls from Megalodon on the individual reads. We found that the measurements were not aligned as well as those from nanoHiMe and CUT&Tag-BS (Fig. 3d, right panel, Pearson's $r = 0.77$).

## nanoHiMe-seq sensitively profiles chromatin features at low coverage depths

Nanopore-based methods have been shown to precisely call DNA methylation at low coverage depths[22]. To assess the minimum sequencing depth required by nanoHiMe-seq to measure CpG methylation and to profile histone modifications, we subsampled the nanopore reads with coverage depths ranging from 5× to 50×. We calculated the methylation percentages of CpG sites based on nano-HiMe calls on the sampled reads and compared the measurements obtained from different coverage depths to the methylation levels determined by WGBS (average coverage >100). Overall, the methylation levels measured from each coverage depth showed a high correlation with the measurements from WGBS (Fig. 4a, b). The correlation coefficient was as high as 0.82 at 10× coverage, corresponding to the reads produced by only one or two MinION flowcells from 100, 000 to 200, 000 cells, and almost reached a plateau (~0.92) at 30× coverage (Fig. 4b). As expected, the methylation levels measured at 10× coverage also correlated well with those measured at 50× coverage (Fig. 4c, Pearson's $r = 0.914$). When analyzing the enrichment of 6mA signals from the sampled reads, we observed that the patterns resembled the profiles generated by CUT&Tag and ChIP-seq, even at a 5× coverage depth (Fig. 4a). Moreover, the 6mA signals detected at 30× or 10× coverage showed a high correlation with those from 50× coverage depth (Fig. 4d). These results suggest that nanoHiMe-seq is a sensitive method that can be used to concurrently measure histone modifications and DNA methylation at a low cost.

## nanoHiMe-seq sufficiently phases histone modifications and DNA methylation and probes their intrinsic connectivity

Nanopore sequencing reads span over several kilobases, which makes them ideal for phasing 6mA-contaning sites and methylated and unmethylated CpGs to their respective haplotypes, especially at the

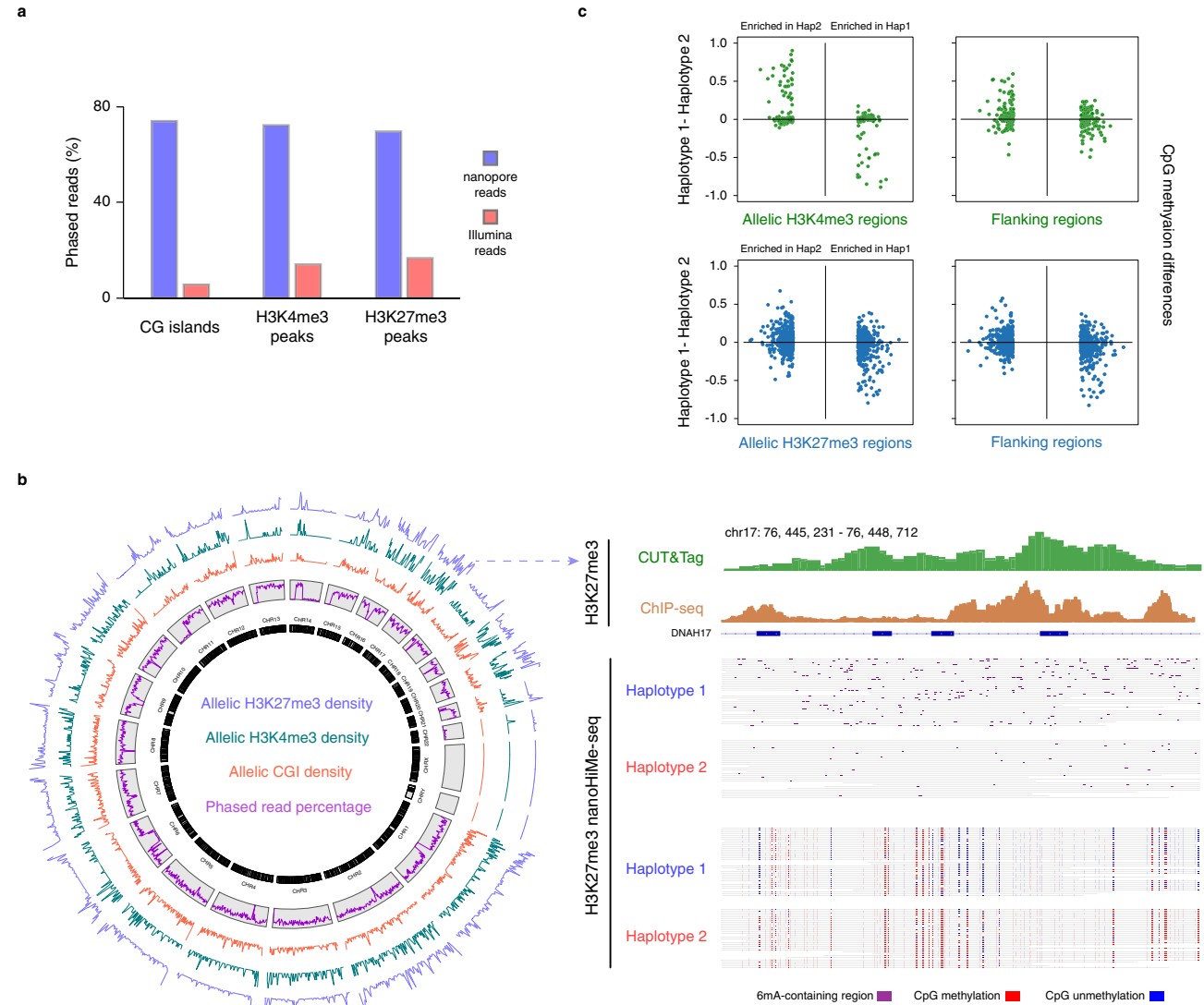

**Fig. 5 | Identification of the heterogeneity and connectivity of H3K4me3, H3K27me3 and CpG methylation. a** Histogram showing the percentages of phased nanopore (blue) and Illumina (red) reads overlapping CpG islands (CGIs) and H3K4me3- and H3K27me3-marked regions. **b** Genomic distribution of allelic CGIs, H3K4me3 and H3K27me3 in HepG2 cells. Left: circos diagram of the genome showing the density of allelic CGIs (red), H3K4me3 (cyan) and H3K27me3 (blue) in moving 2 Mb windows (arbitrary scale). The percentages of nanopore reads (purple) that could be phased in moving windows is also shown inside gray boxes. Right: Example of a genomic region exhibiting allele-specific H3K27me3 and CpG methylation. Top panel shows H3K27me3 signals from CUT&Tag (green) and ChIP-seq (brown); bottom panel shows the identified 6mA-containing sites (purple), methylated CpG sites (red), and unmethylated CpG sites (blue) in individual phased nanopore reads. **c** CpG methylation differences between two haplotypes in and around H3K4me3 or H3K27me3 peak regions with allele-specific enrichment of 6mA signals. Top left panel shows the CpG methylation differences in H3K4me3 peak regions (green), which are categorized into groups with 6mA signals enriched in haplotype 1 (green dots on the right-hand side) or haplotype 2 (green dots on the left-hand side). Top right panel shows the CpG methylation differences in 500-bp flanking regions (green). Bottom two panels show the CpG methylation differences between two haplotypes in H3K27me3-marked regions (left, blue) and ± 500-bp flanking regions (right, blue). Source data are available in the Supplementary Data file 5–7.

genomic loci with a low density of heterozygous single-nucleotide variants (SNVs). To identify allele-specific epigenetic states across the genome, we first used PEPPER-Margin-DeepVariant to find SNVs and insertions and deletions (INDELs) from the reads aligned to the reference[23]. These identified SNVs and INDELs were subsequently used to phase and haplotag the individual nanopore reads (Supplementary Data 4). We found that 72.2% of nanopore reads overlapping H3K4me3 peaks, 69.6% of reads overlapping H3K27me3 peaks and 74.1% of reads overlapping CpG islands (CGIs) were assigned to their respective alleles, whereas only 14.1% of H3K4me3 CUT&Tag reads, 16.6% of H3K27me3 CUT&Tag reads and 5.7% of WGBS reads were phased (Fig. 5a). We grouped these phased nanopores reads from HepG2 cells for individual regions, evaluated CpG methylation levels for every

haplotype, and identified 1236 CGIs exhibiting allele-specific CpG methylation, of which only approximately 30% had been documented in previous studies[14] (Fig. 5b; Supplementary Fig. 8a; Supplementary Data 5). More importantly, we found 800 H3K27me3-marked regions and 476 H3K4me3-marked regions showing allele-specific enrichment of 6mA signals (Fig. 5b; Supplementary Fig. 8b–e; Supplementary Data 6, 7). Very few of these regions had been reported to show allele-specific H3K27me3 or H3K4me3 in previous genome-wide studies due to the technical limitations[18,24]. Moreover, in GM12878 cells, we found 1292 CGIs showing allele-specific CpG methylation, one-third of which were CGIs in X-chromosome, and 1,416 H3K27me3-marked regions exhibiting allele-specific enrichment of 6mA signals (Supplementary Fig. 7; Supplementary Data 5, 6).

nanoHiMe-seq also enabled us to investigate the crosstalk between histone modifications and DNA methylation at the single molecule level along multikilobase segments of the genome. We focused on H3K4me3- and H3K27me3-marked regions that exhibited allele-specific enrichment of 6mA signals and their flanking regions (±1 kb). We investigated CpG methylation on the grouped reads for every haplotype in selected regions in HepG2 cells and found that, in contrast to high methylation levels in the flanking regions, most of CpG sites overlapping the H3K4me3 peaks tended to be unmethylated and hence did not exhibit allele-specific CpG methylation (Supplementary Fig. 9a). In contrast, we observed that 74 H3K4me3-marked regions showed allele-specific CpG methylation and the 6mA signals were preferentially enriched on the opposite allele of the CpG methylation sites (Fig. 5c; Supplementary Fig. 9a). When investigating the methylation levels of their flanking regions, we found 45 regions within 500 bp and 32 regions within 1000 bp also exhibited allele-specific CpG methylation (Supplementary Fig. 9a). Additionally, we found that 95 H3K27me3-marked regions and the majority of their flanking regions (59/95 within 500 bp, 56/95 within 1 kb) exhibited allele-specific CpG methylation in HepG2 cells (Supplementary Fig. 9b). In GM12878 cells, 84 H3K27me3-marked regions and about one-third of their flanking regions (30/84 within 500 bp, 36/84 within 1 kb) showed allele-specific CpG methylation. The 6mA signals were enriched on the allele with the CpG methylation in some H3k27me3-marked regions, but on the opposite allele in other peak regions (Fig. 5c; Supplementary Fig. 9b), which was consistent with the previous observations of the complex and highly dynamic relationship between DNA methylation and H3K27 trimethylation[25].

## Discussion

Here, we describe the nanoHiMe-seq method for simultaneous profiling of histone modifications and DNA methylation via nanopore sequencing. Like CUT&Tag, the workflow of nanoHiMe-seq is quite simple and the entire procedure from cell harvest to library preparation can be performed in 1 day. nanoHiMe-seq is an antibody-based method, can take advantage of the large quantity of antibodies developed for CUT&Tag and ChIP-seq, and be potentially applied to any epitope on chromatin. More importantly, nanoHiMe-seq is a robust and sensitive method that can be used to profile chromatin features using sequencing reads produced by one or two MinION flowcells. The ease, robustness, and low cost of nanoHiMe-seq make it an appealing approach for studying the functional coordination of epigenetics in diverse areas of biological research. In addition, the long reads and no-amplification strategy also make nanoHiMe-seq suitable for the investigation of chromatin features in complex genomic regions and for phasing the features of interest on a genome-wide scale.

In contrast to BIND&MODIFY, which uses methyltransferase Eco-GII to mark adenines, nanoHiMe-seq used Hia5, a non-specific methyltransferase with higher enzymatic efficiency, to label adenine bases proximal to H3K27me3 or H3K4me3 nucleosomes[26]. We also developed a computational tool to simultaneously identify 6mAs and the endogenous mCpGs from individual nanopore reads using an HMM. Systematic benchmarking of methylation prediction by nano-HiMe and Megalodon, a tool provided by ONT, indicated that both tools precisely identified mCpG- and 6mA-containing sites from DNA with one type of modification. However, while nanoHiMe also showed high performance in predicting methylations from DNA containing both mCpG and 6mA, Megalodon couldn't accurately identify methylated CpG sites, and its accuracy for predicting 6mA-containing sites was also significantly lower than the accuracy of nanoHiMe. The robustness of nanoHiMe to call CpG and adenine methylation may result from (1) the trained parameters of emission distribution for individual $k$-mers from DNA with mCpG, 6mA, both mCpG and 6mA, or free of methylation, and (2) the consideration and realignment of

sequential current events to the reference substrings using the trained parameters when evaluating a CpG(s)- and/or adenine(s)-containing site.

CUT&Tag-BS is a recently developed method that has been used to profile the genomic localization of histone modifications and the methylation status of the underlying DNA in a single assay[7]. When performing CUT&Tag-BS experiments, we noticed that extensive efforts were required to optimize the bisulfite treatment conditions. While harsh conditions destroy the limited amount of DNA fragments obtained from CUT&Tag, mild conditions result in incomplete conversion of unmethylated cytosines to uracils. When comparing CUT&Tag-BS to CUT&Tag, we found that, although the peaks detected by CUT&Tag-BS largely overlapped with those identified in CUT&Tag experiments, some of the weak peaks identified by CUT&Tag were not detected by CUT&Tag-BS, probably due to the DNA degradation resulting from bisulfite treatment, even under mild conditions. When comparing nanoHiMe-seq with CUT&Tag-BS, we observed that the 6mA signals were enriched not only at the CUT&Tag-BS peak regions, but also at the regions of weak peaks identified only by CUT&Tag, thus highlighting the high sensitivity of nanoHiMe-seq. We also found that the methylation levels of individual CpG sites measured by nanoHiMe-seq and CUT&Tag-BS were highly correlated. It is worth noting that, unlike CUT&Tag-BS, nanoHiMe-seq also measures methylation levels of the CpGs around the regions marked by modified nucleosomes of interest, enabling the investigation of the relationship between histone modification and methylation of the CpGs inside and outside of the peak regions. Moreover, the long reads obtained from nanoHiMe-seq also allowed us to phase 6mA-contaning regions and methylated and unmethylated CpGs to their respective haplotypes, especially at the genomic loci with a low density of heterozygous SNVs and/or INDELs.

It has been shown that modulation of the current through the nanopore allows the discrimination of many types of base modifications[11,13,16,27]. Thus, nanoHiMe-seq may be scaled up to simultaneously profile multiple chromatin features and probe their intrinsic connectivity by introducing various labels, such as 5-hmC, 5-fC and 4-mC, and GpC methylation. To explore these marks in single DNA molecules, more-extensive training models for all of the combinations of the introduced modifications are required. In our future work, we will aim to generate a such unified calling model that allows for comprehensive epigenetic profiling in a single assay.

## Methods

### pA-Hia5 and pA-Tn5 protein production

We followed Yin et al.[28] to clone, express and purify pA-Hia5 and pA-Tn5 protein from *E. coli* cells. pA-Hia5 protein expression vectors having a 3× Flag-pA-Tn5-Fl backbone (addgene, Plasmid #124601) were adapted by replacing Tn5 coding region with Hia5 gene. 3× Flag-pA-Hia5-Fl and 3× Flag-pA-Tn5-Fl plasmids were transformed into Rosetta 2(DE3)pLysS Competent Cells (Sigma, 71403-3), respectively, following the manufacturer's protocol. Each selected colonies were inoculated into 50 mL LB medium supplemented with 50 μg/ml carbenicillin and 30 μg/ml chloramphenicol, and grown overnight at 37 °C. Overnight culture was added in 1:40 ratio into 2 L auto-inducing ZYP5052 medium, and grown for 4 h at 37 °C to followed by protein expression for 8 h at 16 °C. Bacteria were pelleted at 3700 g at 4 °C for 15 min and resuspended in 200 ml chilled EDTA-free HEGX buffer (20 mM HEPES-KOH at pH 7.2, 0.8 M NaCl, 10% glycerol, 0.2% Triton X-100) containing 0.5 mg/ml lysozyme (sigma, L1667) and 2 mM PMSF. The resuspended bacteria were frozen at −80 °C for at least 0.5 h, and protein purification was performed according to Kaya-Okur et al.[5] with minor modifications. Briefly, frozen bacterial lysate was thawed at room temperature and DNA was digested with 10 μg/ml Deoxyribonuclease I (sigma, D2821-50KU) and 3 mM MgSO₄ at room temperature for 30 min. Chitin resin (NEB, S6651S) was prepared by washing 10 ml slurry with 100 ml of EDTA-free HEGX Buffer, and then the soluble

fraction of the lysate was added to the chitin resin slowly. The clarified lysate was incubated with the chitin resin on an end-over-end rotator at 4 °C for 6 h, after which the solution was removed by centrifugation at 1000 $g$ at 4 °C for 5 min. The chitin resin was washed three times with chilled HEGX buffer, resuspended in 40 mL HEGX including 100 mM DTT, and then rotated at 4 °C for about 48 h. 20 K MWCO dialysis cassettes (Thermo Scientific, 66012) were pre-wet in 2× dialysis buffer (100 mM HEPES-KOH pH 7.2, 0.2 M NaCl, 0.2 mM EDTA, 2 mM DTT, 0.2% Triton X-100, 20% Glycerol) for 5 min and the elution was transferred to the cassettes and dialyzed twice in 200× volumes of 2× dialysis buffer overnight at 4 °C. Dialyzed sample was transferred to a 30 K Amicon Ultra-15 tube (Millipore, UFC903024) and centrifuged at 3700 $g$ to concentrate to a volume of <1 ml. The concentrated protein was stored at −80 °C until use.

### In vitro methyltransferase activity assessment

DNA used in the assay was amplified by PCR from synthesized ssDNA (GenScript Biotech, Supplementary Data 1), or from HepG2 genomic DNA with primers for a 814-bp region, which located in promoter of the hydroxymethylbilane synthase gene and contained four GATC sequence. Approximately 1 µg DNA was incubated with 0, 0.5, 1, 2, 5, 8 or 10 µl pA-Hia5 protein in 1× methyltransferase reaction buffer (15 mM HEPES-KOH at pH 8.0, 15 mM NaCl, 60 mM KCl, 1 mM EDTA pH 8.0, 0.5 mM EGTA pH 8.0, 0.5 mM Spermidine) supplemented with 160 µM S-adenosyl-methionine (NEB, B9003S) at 37 °C for 2 h. After cleaned up using MinElute PCR purification kit (Qiagen, 28004), 100 ng DNA was subjected to DpnI (ThermoFisher Scientific, FD1704) or MboI (ThermoFisher Scientific, FD0814) digestion followed the manufacturer's protocol. 50 ng DNA from each reaction was combined with 2 µl of 6× Purple Gel Loading Dye (NEB, B7024) and ran on a 2% agarose gel supplemented with 1× SYBR Safe DNA Gel Stain (thermo, S33102) at 120 V for approximately 40 min. The gel was imaged on ChemiDoc™ Touch Imaging System and the image was cropped using Image Lab v5.2.1. To evaluate whether the methyltransferase activity of Hia5 was affected by CpG methylation or not, 1 µg amplicon from synthesized ssDNA (Supplementary Data 1) was either incubated with 4 units CpG Methyltransferase M.SssI (NEB, M0226L) in NEBuffer 2 (NEB, B7002) containing 160 µM S-adenosyl-methionine or incubated with 15 µl pA-Hia5 protein in 1× methyltransferase reaction buffer supplemented with 160 µM S-adenosyl-methionine at 37 °C for 2 h and then purified using MinElute PCR purification kit, respectively. Approximately 1 µg M.SssI-treated DNA was then mixed with 15 µl pA-Hia5 protein in 1× methyltransferase reaction buffer supplemented with 160 µM S-adenosyl-methionine and the incubation was performed under 37 °C for 2 h. The amplicon, M.SssI-treated DNA, Hia5-treated DNA, and M.SssI&Hia5-treated DNA was subjected to DpnII (NEB, R0543S) or PvuI-HF (NEB, R3150S) digestion according to the manufacturer's protocol. 50 ng DNA from each reaction was ran on a 4% agarose gel and the gel was imaged as described above.

### Cell culture

HepG2 cells were obtained from the American Type Culture Collection (ATCC, HB-8065) and grown at 37 °C in a humidified incubator in 1× EMEM medium (ATCC, 30-2003) supplemented with 10% FBS (ThermoFisher Scientific, 10100147c) and 1% Pen Strep (ThermoFisher Scientific, 15140122). GM12878 cell line was kindly donated by Wensheng Wei (School of Life Sciences, Peking University) and cultured in RPMI 1640 medium (Gibco 11875-093) supplemented with 15% FBS and 1% Pen Strep. To deplete H3K27 methylation, GM12878 cells were treated with high concentrations (2 or 5 µM) of the EPZ6438 EZH2 inhibitor (Cayman Chemical, 16174) for 7 days[19].

### Extraction of nucleosomal histones

We basically followed the high-salt extraction of histones protocol described by Shechter et al.[29]. Briefly, $1 \times 10^5$ GM12878 cells were resuspended in 1 ml extraction buffer (10 mM HEPES pH 7.9, 10 mM KCl, 1.5 mM MgCl$_2$, 0.34 M sucrose, 10% glycerol, 1× complete, EDTA-free protease inhibitor cocktail) with 0.2% IGEPAL CA-630 (sigma, I8896) and incubated for 10 min on ice with occasionally rotating. The nuclei were collected by centrifugation at 6500 g at 4 °C for 5 min and washed once with 1 ml extraction buffer. The nuclei were lysed in 1 ml no-salt buffer (3 mM EDTA, pH8.0) for 30 min at 4 °C and the chromatin pellet was collected by centrifugation at 6500 $g$ at 4 °C for 5 min. The chromatin pellet was resuspended in 600 µl high-salt solubilization buffer (50 mM Tris-Cl pH 8.0, 2.5 M NaCl, 0.05% IGEPAL CA-630) and incubated on a rotator at 4 °C for 30 min. After centrifugation at 16,000 $g$ at 4 °C for 10 min, the supernatant was transferred into the cassettes (3.5 K MWCO) and dialyzed in dialysis buffer (10 mM Tris-Cl pH 8.0, chilled to 4 °C) overnight at 4 °C. Finally, the histone solution was collected and stored at −80 °C freezer. H3 and H3K27me3 proteins were detected by western blotting.

### CUT&Tag and CUT&Tag-BS

We basically followed the CUT&Tag protocol described by Kaya-Okur et al.[5]. Briefly, nuclei of HepG2 and GM12878 cells were prepared by suspending and incubating cells in chilled NE1 for 10 min. Nuclei were resuspended in PBS, lightly cross-linked using 0.1% fomaldehyde for 2 min and then the cross-linking was stopped by 75 mM glycine. Fixed nuclei were collected by centrifugation at 1300 g for 4 min at 4 °C, resuspended in wash buffer to a concentration of 1 million per 1 ml and then bound to ConA magnetic beads. ConA-bound nuclei were incubated with non-specific IgG (abcam, ab46540) or antibody against H3K4me3 (Active Motif, 39159) or H3K27me3 (Cell Signaling Technology, 9733 S) in wash buffer containing 2 mM EDTA, 0.1% BSA (1:50 dilution) for 2 h at 25 °C and then incubated with secondary antibody (antibodies online, ABIN101961) in wash buffer (1:100 dilution) for 30 min 25 °C. The nuclei were washed once using wash buffer and then incubated with 0.4 µl pA-Tn5 (Diagenode, C01070001) in 50 µl wash buffer containing 300 mM NaCl for 1 h at 25 °C. After being washed 3 times using wash buffer containing 300 mM NaCl, the nuclei were brought up in 300 µl wash buffer containing 300 mM NaCl and 10 mM MgCl$_2$, and then incubated at 37 °C for 1 h to allow the tagmentation reaction to go to completion. DNA was extracted and PCR reaction was performed according to the protocol provided by Kaya-Okur et al.[5]. The PCR products were subjected to Illumina sequencing (2 × 150 bp, adapters and amplification primers listed in Supplementary Data 1).

CUT&Tag-BS protocol was similar to CUT&Tag protocol, but had two extra steps, oligonucleotide replacement and bisulfite treatment. As described by Li et al.[7] for oligonucleotide replacement and gap repair, 11 µl CUT&Tag DNA, 2 µl 10 µM Tn5mC-ReplO1 oligo (Supplementary Data 1), 2 µl 10× Ampligase buffer (Lucigen, A3202K) and 0.5 µl dNTP mix (10 mM each) were mixed and incubated in PCR thermocycler as follows: 50 °C for 1 min, 45 °C for 10 min and ramp down to 37 °C at a rate of −0.1 °C/s. 1 µl T4 DNA polymerase (NEB, M0203S) and 2.5 µl Ampligase (Lucigen, A3202K) were added to the reaction, which was incubated at 37 °C for 30 min. The reaction was stopped by the addition of 1 µl 0.5 M EDTA (pH 8.0) and the repaired DNA was purified using MinElute PCR purification kit. The purified DNA was mixed with 130 µl of the CT conversion reagent from EZ DNA Methylation-Gold Kit (Zymo Research, D5005). The solution was incubated in a thermocycler as follows: 98 °C for 8 min, 54 °C for 60 min, hold at 4 °C. The bisulfite converted DNA was purified according to the instructions from EZ DNA Methylation-Gold Kit and eluted in 25 µl M-elution buffer. PCR reaction was performed according to the protocol described by Kaya-Okur et al.[5].

### CUT&Tag and CUT&Tag-BS data processing

The quality of sequencing reads from CUT&Tag experiments was viewed using FastQC v0.11.5 and the reads were aligned to human reference genome HG19 using Bowtie2 (version 2.4.4)[30] with

parameters --end-to-end --very-sensitive --no-mixed --no-discordant --phred33 -I 10 -X 700. All unmapped reads, non-uniquely mapped reads and PCR duplicates were removed using Samtools v1.9, and H3K4me3 and H3K27me3 peaks were called using MACS2 v2.2.7.1[31] with the parameters -broad -g hs -f BAMPE.

CUT&Tag BS-seq reads were aligned to human reference genome HG19 using Bismark (version −0.22.3) with Bowtie2 (version2.4.4) as the alignment software. PCR duplicates were removed using deduplicate_bismark. Per-residue methylation data was collected using bismark_methylation_extractor with parameters -comprehensive --bedGraph --counts --cytosine_report. HOMER v4.11[32] was used to call peaks as follows: Mapped read data was prepared using the make-TagDirectory function with parameters "-read1-keepAll -fragLength 200", followed by peak calls using the findPeaks function with parameters "-size 1000 -minDist 2500 -L 0 -region". We used bedtools v2.25.0 to identify common peaks between CUT&Tag-BS and CUT&Tag, and between biological replicates from CUT&Tag-BS or CUT&Tag. The enrichments of H3K27me3 and H3K4me3 by CUT&Tag or CUT&Tag-BS across different segments of the human genome were viewed using IGV v2.9.2, and heatmaps showing the enrichment signals of H3K27me3 and H3K4me3 from CUT&Tag were generated using deeptools v3.5.1.

### Comparison between nanoHiMe-seq, CUT&Tag and ChIP-seq or between nanoHiMe-seq and WGBS

The HepG2 ChIP-seq data used in this study was from Encyclopedia of DNA Elements (ENCODE) with accession number ENCFF223BZS, ENCFF632WJV and ENCFF712HMU for H3K4me3 peak information, ENCFF942HPS, ENCFF833HZR, ENCFF447EJD, ENCFF663ZBD and ENCFF053ZWM for alignment information of individual sequencing reads from H3K4me3 ChIP-seq experiments; ENCFF546TJV and ENCFF950VUB for H3K27me3 peak information; ENCFF591ZGI, ENCFF502ALT, ENCFF730WFM and ENCFF560BZE for alignment information of individual sequencing reads from H3K27me3 ChIP-seq experiments. As almost all of the top 50% peaks from ChIP-seq overlapped with those identified by CUT&Tag and vice versa, so these peaks represented the high-confidence H3K27me3- or H3K4me3-marked regions and were selected for the comparison analysis. The correlation between nanoHiMe-seq, CUT&Tag and ChIP-seq or between the replicates of CUT&Tag and ChIP-seq was calculated using either the read counts or 6mA-containing sites in the selected peak regions.

Whole-genome Bisulfite Sequencing (WGBS) data was from ENCODE with accession numbers ENCFF847OWL, ENCFF390OZB, ENCFF064GJQ and ENCFF369YQW. The correlation between nanoHiMe-seq and WGBS was calculated for the CpG sites with coverage depth ≥20 by both WGBS and nanoHiMe-seq data. The active promoters used in Supplementary Fig. 5 were selected based on HepG2 RNA-seq data from ENCODE under accession numbers ENCSR000CPC, ENCSR000CPE and ENCSR000CPF.

### nanoHiMe-seq using antibody against H3K4me3, H3K27me3 or IgG

HepG2 cells were harvested by trypsinization upon reaching approximately 60% confluency in 100 mm dishes (Corning, 430167) and GM12878 cells were harvested by centrifugation for 4 min at 600 g at room temperature. Cells were washed once in PBS and then resuspended in ½ volume (relative to cell culture medium) chilled NE1 buffer (20 mM HEPES-KOH at pH 7.9, 0.5 mM spermidine, 0.1% Triton X-100, 20% glycerol) containing 1× Complete, EDTA-free Protease Inhibitor Cocktail (Roche 5056489001). Cells were lysed for 10 min on ice and then the nuclei were pelleted by centrifugation at 1300 g for 4 min at 4 °C. The nuclei were resuspended in wash buffer (20 mM HEPES pH 7.5, 150 mM NaCl, 0.5 mM Spermidine, 1× Complete, EDTA-free Protease Inhibitor Cocktail) to a concentration of

approximately 1 million per ml. Magnetic beads coated by Concanavalin A (Bangs Laboratories, BP531) were prepared as described by Kaya-Okur et al.[5] and 4.5 µl of activated beads were added for 100,000 nuclei and incubated at room temperature for 10 min. The supernatant was removed after placing the EP-tubes containing magnetic beads and nuclei on the magnet stand for 1 min and the nuclei were resuspended in 50 µl wash buffer containing 2 mM EDTA, 0.1% BSA and 1 µl antibody (1:50 dilution) against H3K4me3 (Active Motif, 39159), H3K27me3 (Cell Signaling Technology, 9733 S) or non-specific IgG (abcam, ab46540). The nuclei were incubated with primary antibody for 2 h at 25 °C on ThermoMixer C with mixing frequency at 1600 rpm. The primary antibody was removed by pulling off the liquid from the tubes after placing them on magnet stand for 1 min. The nuclei were resuspended in 50 µl wash buffer supplemented with 0.5 µl Guinea Pig anti-Rabbit secondary antibody (antibodies online, ABIN101961, 1:100 dilution) and incubated for 30 min at 25 °C on ThermoMixer C. After removing secondary antibody, the nuclei were washed once using 500 µl wash buffer and then resuspended in 50 µl wash buffer containing 300 mM NaCl and 3 µl pA-Hia5. After incubation with pA-Hia5 protein for 1 h, the nuclei were washed 3 times using wash buffer with 300 mM NaCl and then resuspended in 1 × methyltransferase reaction buffer supplemented with 160 µM S-adenosyl-methionine. The reactions were incubated at 37 °C for 2 h and then for an additional 2 h at 55 °C after adding 0.5 µl of 10% SDS, 1.7 µl 0.5 M EDTA and 1 µl 20 mg/ml proteinase K (ThermoFisher Scientific, 25530049) to each sample. All samples were mixed with 0.5 µl RNase A (ThermoFisher Scientific, EN0531) and incubated for 30 min at 37 °C. After pooling all samples together, the DNA was purified using Nanobind CBB Big DNA Kit (Circulomics, SKU NB-900-001-01) according to the manufacturer's protocol.

### Data generation for training model

Genomic DNA from HepG2 cells and from *E. coli* K12 MG1655 (ATCC, 47076) was extracted using Nanobind CBB Big DNA Kit (Circulomics, SKU NB-900-001-01). Purified genomic DNA was sheared to a fragment of ~10 kb using Covaris g-TUBE (Covaris, 520079), and the fragmented DNA was end-repaired and dA-tailed by combining ~1 µg DNA with NEBNext FFPE DNA Repair Mix (NEB, M6630S) and Ultra II End-prep/dA-Tailing enzyme mix (NEB, E7546S). Samples were mixed, placed on a thermo cycler using the end repair program (5 min at 20 °C, 5 min at 65 °C, then hold at 4 °C) and then cleaned up using 1× v/v AMPure XP beads (Beckman Coulter, A63881). PCR adapters were ligated to the fragmented DNA using NEB Blunt/TA Ligase followed by 14 cycles of PCR amplification according to Nanopore Protocol of PCR Sequencing Kit (ONT, SQK-PSK004). The experimental procedure from Lee et al. was followed in order to obtain DNA samples with near-complete CpG methylation using M. SssI enzyme (NEB, M0226L)[13]. In contrast to near-complete CpG methylation, we intentionally obtained DNA samples with partially methylated adenines using home-made pA-Hia5 protein, as well as DNA samples with near-complete CpG methylation and partial adenine methylation using pA-Hia5 or/and M. SssI enzyme: First, ~1 µg amplicons were either incubated with 10 µl pA-Hia5 in 1× methyltransferase reaction buffer supplemented with 160 µM S-adenosyl-methionine, or incubated with 8 units M. SssI in NEBuffer 2 supplemented with 160 µM S-adenosyl-methionine for 2 h at 37 °C. Second, after being cleaned up using MinElute PCR purification, ~1 µg amplicons with methylated adenine were re-incubated with 10 µl pA-Hia5 and that with methylated CpG were re-incubated with 8 units M. SssI for an additional 2 h at 37 °C. Last, ~1 µg amplicons with 2 cycles of M. SssI treatment were incubated with 10 µl pA-Hia5 for an additional 2 cycles as described above. The untreated amplicons, amplicons with 2 cycles of adenine methylation, 2 cycles of CpG methylation, or 2 cycles of CpG methylation + 2 cycles of adenine methylation were subjected to nanopore sequencing.

## Nanopore sequencing

The nanopore sequencing library from purified genomic DNA was prepared according to the protocol in the genomic DNA by ligation kit (ONT, SQK-LSK109). First, approximately 3 µg genomic DNA was sheared using g-TUBE by centrifugation at 3400 g (eppendorf centrifuge 5424 R) for 1 min and then an additional 1 min after inverting the tube to achieve a fragment length of ~10 kb. The fragmented DNA was repaired and dA-tailed as described above. Sequencing adaptors bound by motor proteins were ligated to the repaired and dA-tailed DNA using Quick T4 DNA Ligase (NEB, E6056S), and then the DNA together with bound protein were pulled down by 0.4× v/v AMPure XP beads and cleaned up 2 times using Long Fragment Buffer (LFB) from sequencing kit. ~400 ng adaptor-ligated DNA was loaded onto each FLO-MIN106D flow cell (ONT, 11001832), which was run on MinION Mk1b sequencer for 72 h. Data were collected by MinKNOW v4.1.22.

## Nanopore sequencing data preprocessing

Basecalling was performed using Guppy v4.4.2 with the "high accuracy" model. Individual reads were aligned to hg19 human reference genome by Minimap2 v2.17 with default settings[33]. The breadth and depth of nanoHiMe-seq coverge for H3K27me3 and H3K4me3 were presented in Supplementary Fig. 4c and Supplementary Data 2. The alignment of the electric current events to k-mers of a known substring from the reference genome was obtained using nanopolish v0.13.2. For training and testing data, the event alignment was further optimized using Viterbi algorithm (see Model training for details).

## Training of model parameters using SignalAlign v0.3.0

SignalAlign model training was performed using script "trainModels.py" with the following settings: probability_threshold: 0.8, number_of_kmer_assignments: 10, training_bases: 10000, transitions: false, normal_emissions: true, hdp_emissions: true, expectation_maximization: false, em_iterations: 2. 10,000 ONT sequencing reads from each of PCR+ M.SssI+ Hia5-, PCR+ M.SssI- Hia5+ and PCR+ M.SssI+ Hia5+ datasets were randomly selected as input data for training of model parameters. The output "hmm models" were used as the "template_hmm_model" for next iteration. The iteration stoped when the parameters of individual k-mers in "hmm models" became constant.

## Haplotype assignment and allele-specific analysis of DNA methylation, H3K4me3 and H3K27me3

HG002 nanopore fast5 files (https://s3-us-west-2.amazonaws.com/human-pangenomics/index.html?prefix=NHGRI_UCSC_panel/HG002/nanopore/) were base-called using Guppy v4.4.2 with the "high accuracy" model. Individual reads were aligned to human reference genome HG19 by Minimap2 v2.17 with default settings and the mapped reads were used to re-train PEPPER and DeepVariant models by following the instructions. We used PEPPER-Margin-DeepVariant (version 0.8.0) with the trained models to identify single-nucleotide polymorphisms (SNPs) and INDELs from the nanoHiMe-seq reads alignments to the hg19 reference. We used hap.py version v0.3.12 to assess the variant calls against GIAB truth set (NIST v3.3.2/GRCh37, GM12878 cell) or the dataset (HepG2 cell) obtained by Zhou et al.[18]. The resulting phase block lengths, switch error rate and hamming error rate were listed in Supplementary Data 4. We used Margin version v2.2 and WhatsHap version v1.4 to haplotag and phase the variants. The haplotagged nanopore reads were grouped in individual CGIs, H3K4me3 or H3K27me3 peaks and the regions with reads coverage ≥20 were selected for the following analysis. The number of CpG methylation calls and unmethylation calls on grouped reads in each CGI was recorded and used to evaluate allele-specific methylation through Fisher's exact test. In addition, the grouped reads overlapping H3K4me3 or H3K27me3 peak regions were segmented into non-overlapping 250 bp windows, and the windows located in each peak region were categorized into two groups: the ones harboring 6mA(s) and the ones don't. A bootstrap procedure was applied by sampling 30 250-bp-windows for each bootstrap replication from every allele and the number of windows containing 6mA(s) was recorded. The bootstrapped distribution was generated by considering 100 bootstrap replications and two-sided Fisher's exact test was used to evaluate allele-specific enrichment of 6mA-containing regions. The statistical significance was corrected for multiple hypotheses using Benjamini-Hochberg approach and the statistically significant results ($p < 0.01$) were selected using a target false discovery rate of 10%.

## Classification of methylation sites with Megalodon

Fast5 files were modification basecalled with Megalodon (v2.5.0) using model res_dna_r941_min_modbases-all-context_v001.cfg with --outputs mod_mappings. The output mod_mappings.bam files were used to compute methylation probability of every CpG or A site in each read. When calling adenine methylation, modification basecalled reads were smoothed by calculating rolling average over windows ranging from 12 to 100 bp in a NaN-sensitive manner (averaging only over adenine bases).

## Model training

**hidden Markov model.** As previous studies[10,15,16], a hidden Markov model was used to calculate the probability of observing a sequence of current events $e_1, ..., e_n$ from nanopore sequencing device under the assumption of that the true nucleotides sequence is $S$. A block of states were constructed for every k-mer of $S$, including *match*, *skip*, *bad*, and *softclip* states, and a self-transition was allowed for the *match* state to account for two or multiple observations of events from a particular k-mer. The transition probabilities in our model were also set to constant values and the emission distributions were described in 1.2[5].

In this study, when a sequence of current events, $e_1, ..., e_n$ was observed, the likelihood of the underlying nucleotides sequence to be $S$ was calculated by:

$$\mathcal{L}(S|e_1, ..., e_n, \Theta) = P(e_1, ..., e_n|S, \Theta) \quad (1)$$

Where $\Theta$ is the model parameters of the emission distributions of individual k-mers.

**Emission distributions.** For R9.4 data, we also followed the principles that the probability of observing an event $e_i$ given that the true sequence in the pore is k-mer $k$ is modeled by a Gaussian distribution[10,34]:

$$P(e_i|k, \Theta) = \mathcal{N}\left(a + b\mu_k, (d\sigma_k)^2\right) \quad (2)$$

Here, the equation is the emission distribution for the *match* state of R9 Hidden Markov Model. $a$, $b$ and $d$ are used to define the read-specific deviations from the reference model N ($\mu_k$, $\sigma_k^2$) and calculated using the event labels of reference k-mers through linear least square method.

**Expanding the emission lexicon.** There are 4096 different Gaussian distributions provided by ONT corresponding to individual 6-mers consisted of four standard base nucleotide alphabets (A, C, G, T). In order to handle two types methylation, we expanded the nucleotide alphabet from the four standard nucleotide bases to six nucleotide bases (A, C, G, T, M, Z) with M representing 5-methylcytosine and Z 6-methyladenine, which resulted in an increase in the number of emission distributions to $6^6 = 46656$. Since cytosine methylation only happened in CpG dinucleotides in current study, all of the k-mers with MH (H represents A, C and T) and MZ were invalid methylation sites, such as TAMAGG, TAMCGG, TAMTGG and TAMZGG, whereas TAMGGG was a valid 6-mer. In this study, we intentionally introduced partial adenine

methylation, which enabled us to obtain parameters of $k$-mers with 6mA in all possible contexts. As the partial adenine methylation made it impossible to precisely assign the learned parameters to $k$-mers that were derived from a four-letter $k$-mer, but contain methyladenine(s) in different positions, such as TZCACG, TACZCG and TZCZCG, where Z denotes 6mA. Hence, we replaced such $k$-mers that contain methylated adenine(s) and share the same nucleotide bases composition and order except A/Z by $k$-mer $\bar{k}$. For instance, the 6-mers TZCACG, TACZCG, TZCZCG were replaced by $\overline{\text{TACACG}}$, and TZCAMG, TACZMG, TZCZMG by $\overline{\text{TACAMG}}$.

**Training emission distributions.** The following is the description about how to train new parameters of the emission distributions of individual $k$-mers.

We first performed basecalling using Guppy v4.4.2 with the "high accuracy" model for the output of MinION sequencing device and then aligned individual reads to the reference genome using Minimap2 with default settings[33]. For the reads with mapping quality ≥20, we performed initial alignment between events measured by MinION and $k$-mers of reference subsequence $S$ using nanopolish v0.13.2, after which a collection of events was obtained for each $k$-mer $k$ or $\bar{k}$. The collections of events were used to train new parameters of individual $k$-mers as described by Simpson et al.[10] and the trained new parameters were used for the realignment of events to $k$-mers of substring $S$ using Viterbi algorithm. We iterated the process for 5 times and assigned the parameters from the final iteration to each $k$-mer for the downstream analysis in this study.

As previous study[10], we also used the region from 50,000 bp to 3,250,000 bp of the *E. coli* K12 MG1655 genome for the parameters training. Before training, the initial parameters for unmethylated $k$-mers were set to ONT-provided values. For methylated $k$-mers, standard deviations were increased by 2 while keeping the mean the same as unmethylated version of such $k$-mer.

**Fitting gaussian mixture models.** Let $E_k$ denote the sequence of events aligned to a $k$-mer, and $E'_k$ be the sequence of transformed events to accounts for shift and scale parameters, where $E_k = (e_1, \ldots, e_n)$, $E'_k = (e'_1, \ldots, e'_n)$, and $e'_i = \frac{e_i - a_i}{b_i}$. After transformation, the distribution of $e'_i$ follows a single gaussian model with $P(e'_i) = \mathcal{N}(\mu_k, ((d_i/b_i)\sigma)^2)$ in absence of mCpG and 6mA. For the training dataset treated by M. SssI, which methylated nearly all cytosines in a CpG context (96.2%), a two-component Gaussian mixture model was fitted to $E'_k$ when $k$-mer $k$ contains CpG dinucleotides; For the training dataset obtained from pA-Hia5 or pA-Hia5 & M. SssI-treated DNA sample, where partially methylated adenine in various position(s) was introduced, multiple-component Gaussian mixture models were applied to $E'_k$, when $k$-mers are $\bar{k}$ s or contain two types of methylations (M + Z); In contrast, a two-component Gaussian mixture model was fitted to $E'_k$ when $k$-mer $k$ contains only one adenine.

$$P(e'_i) = \sum_j \omega_j \mathcal{N}(\mu_j, ((d_i/b_i)\sigma_j)^2) \tag{3}$$

$$\sum_j \omega_j = 1$$

In the equation, the weight $\omega_j$ either represents the proportion of events coming from methylated or unmethylated sequence in two-component mixture models, or represents the proportion of events coming from a specific modification state of a sequence in multiple-component mixture models; $\mu_j$ and $\sigma_j$ are either the parameters of Gaussian distributions for unmethylated or methylated version of $k$-mer $k$, or the parameters of Gaussian distributions for $k$-mers that constitute $\bar{k}$.

To use the expectation-maximization algorithm to fit $\omega_j$, $\mu_j$, $\sigma_j$, first, we took $\mu_k$, $\mu_k \pm 5$, $\mu_k \pm 10$, $\sigma_k$, $\sigma_k + 1$ as the initial guess of the parameters, where $\mu_k$ and $\sigma_k$ are the parameters of unmethylation version of $k$-mer $k$ from ONT models, and $0.95 + 0.05$, $0.5 + 0.3 + 0.2$ or $0.25 + 0.25 + 0.25 + 0.25$ as the initial value of $\omega_j$. Second, we calculated the *responsibility* of the $j$th component of the mixture model for observing events $e_i$:

*Expectation step*

$$\hat{\gamma}_{i,j} = \frac{\omega_j \phi_{\hat{\Theta}_j}(e_i)}{\sum_j \omega_j \phi_{\hat{\Theta}_j}(e_i)} \tag{4}$$

Last, we updated the estimates for weights and parameters as below:

*Maximization step*

$$\hat{\omega}_j = \frac{\sum_{i=1}^n \hat{\gamma}_{i,j}}{n} \tag{5}$$

$$\hat{\mu}_j = \frac{\sum_{i=1}^n \hat{\gamma}_{i,j} e'_i}{\sum_{i=1}^n \hat{\gamma}_{i,j}} \tag{6}$$

$$\hat{\sigma}_j^2 = \frac{\sum_{i=1}^n \hat{\gamma}_{i,j} \left((e'_i - \mu'_j)\frac{b_i}{d_i}\right)^2}{\sum_{i=1}^n \hat{\gamma}_{i,j}} \tag{7}$$

Different sets of initial parameters were tried for a $k$-mer containing methylated base(s). Expectation step and maximization step were iterated until convergence or iterated 1000 times if convergence didn't happen for that set of initial parameters. For two-component Gaussian mixture model, if two components were finally determined and the parameters were consistent among all or parts of final iterations, the mean with larger differences from ONT model mean and the companioned standard deviation were assigned to $k$-mer $k$ with methylated site(s). For example, if $\hat{\mu}_1$, $\hat{\mu}_2$, $\hat{\sigma}_1$ and $\hat{\sigma}_2$ were consistent among all or parts of the final iterations and $|\hat{\mu}_1 - \mu_{ref}| > |\hat{\mu}_2 - \mu_{ref}|$, then the consistent values of $\hat{\mu}_1$ and $\hat{\sigma}_1$ from the final iteration were assigned to be the parameters of methylated $k$-mer $k$; if only one component was finally determined and the parameters were consistent, then the set of the parameters were assigned to methylated $k$-mer $k$. For multiple-component Gaussian mixture model, the component number was initially guessed based on the distribution of $e'_i$ from a $k$-mer and the parameters from final iterations with mean not equal to the reference mean of unmodified version of $k$-mer $k$ were assigned to $k$-mer $\bar{k}$ or $k$ with two types of methylation. For example, three means and standard deviations ($\hat{\mu}_1, \hat{\mu}_2, \hat{\mu}_3, \hat{\sigma}_1, \hat{\sigma}_2, \hat{\sigma}_3$) were obtained for a $k$-mer $\bar{k}$ from the final iteration. When trying different initial guess for $k$-mer $\bar{k}$, the obtained parameters from the final iterations were consistent. If $\hat{\mu}_1$ was the same with reference mean of unmodified version $k$-mer $k$, then we assigned the values of $\hat{\mu}_2, \hat{\mu}_3, \hat{\sigma}_2$ and $\hat{\sigma}_3$ from the final iteration to $k$-mer $\bar{k}$.

**Classification of methylation sites.** The following is the description about calling CpG methylation and 6mA-containing sites in individual nanopore reads from nanoHiMe-seq using our trained model.

For classification of CpG site(s), we also constructed substring $S_R$, which harbors one or multiple CpG dinucleotides and at least 10 bp CpG free sequences on both sides, and jointly tested CpGs for methylation in each $S_R$ from individual reads[10]. As to adenines, we used 50 bp sliding windows for the classification and, instead of assessing which adenines in a window was methylated or unmethylated, we evaluated whether the window contained methylated adenine(s) or not. We used log likelihood ratio to call a CpG or a group of CpGs as methylated or unmethylated, as well as to call a 50-bp window as a

6mA-containing segment or non-6mA segment. For the substrings containing both adenine(s) and CpG(s), we calculated the likelihood for combinations of being methylated or unmethylated (no methylation - *ref 1*, all CpGs methylated - *ref 2*, 6mA-containing - *ref 3*, all CpGs methylated & 6mA-containing - *ref 4*), using hidden Markov model with the parameters trained above. We calculated a log likelihood ratio for calling CpG methylation by summing the likelihoods of sequences with methylated CpG(s) or unmethylated CpG(s). We did calculation in the same way for calling 6mA-containing sites:

$$
LLR(S_M) = \sum_i \log\left( \max_{\theta \in \Theta} \mathcal{L}(S_i | e_1, \ldots, e_n, \Theta) \right) \\
- \sum_j \log\left( \max_{\theta \in \Theta} \mathcal{L}(S_j | e_1, \ldots, e_n, \Theta) \right)
\tag{8}
$$

Where *i* is *ref 2* and *ref 4*, *j* is *ref 1* and *ref 3* for calling CpG methylation; *i* is *ref 3* and *ref 4*, *j* is *ref 1* and *ref 2* for calling 6mA-containing sites. The set of parameters that resulted in maximum likelihood of observing substrings with methylated CpG(s) and/or 6mA(s) were selected for the calculation when multiple trained sets of parameters were available for a *k*-mer. Throughout the text, we set a threshold of 1.5 to call methylated CpGs and −1.5 to call unmethylated CpGs, and set 8 or 32 to call a 50 bp segment as a 6mA-containing or non-6mA window.

**Exploring H3K4me3/3K27me3 marked regions from nanoHiMe-seq data.** For nanoHiMe-seq data generated using antibody against H3K4me3 or H3K27me3, we used the threshold of 8 or 32 to call 6mA-containing sites in 50 bp windows. The analysis was also performed for the data generated from IgG and the 6mA-containing site called from IgG data were used for background correction in every segment across the genome:

$$R_f = R_s - sf * R_b \tag{9}$$

$$sf = C_s / C_i \tag{10}$$

*sf* means scaling factor; $C_s$ means the coverage depth of the segment by the H3K4me3 or H3K27me3 nanoHiMe-seq data; $C_i$ means the coverage depth of the corresponding segment by IgG data; $R_f$ means number of predicted 6mA-containing sites after background correction; $R_s$ means the number of predicted 6mA-containing sites in H3K4me3 or 3K27me3 nanoHiMe-seq data; $R_b$ means the number of predicted 6mA-containing sites in IgG data.

### Statistics and reproducibility
Unmapped sequencing reads were discarded in analysis, otherwise no data were excluded from the analyses. Two-sided Fisher's exact test was used to evaluate allele-specific enrichment of 6mA-containing regions and allele-specific CpG methylation. The statistical significance was corrected for multiple hypotheses using Benjamini–Hochberg approach and the statistically significant results ($p < 0.01$) were selected using a target false discovery rate of 10%.

### Reporting summary
Further information on research design is available in the Nature Portfolio Reporting Summary linked to this article.

## Data availability
The sequencing data generated in this study have been deposited in the European Nucleotide Archive (ENA) under accession number PRJEB47152. The minimum datasets that are necessary to interpret, verify and extend our research have been deposited in the Zenodo (https://doi.org/10.5281/zenodo.7388709). The relevant processed data are included as Supplementary Information and Source Data. The HepG2 ChIP-seq data used in this study are from Encyclopedia of DNA Elements (ENCODE) with accession number ENCFF223BZS, ENCFF632WJV and ENCFF712HMU for H3K4me3 peak information; ENCFF554AJT, ENCFF942HPS, ENCFF833HZR, ENCFF447EJD, ENCFF663ZBD, and ENCFF053ZWM for alignment information of individual sequencing reads from H3K4me3 ChIP-seq experiments; ENCFF546TJV and ENCFF950VUB for H3K27me3 peak information; ENCFF591ZGI, ENCFF502ALT, ENCFF730WFM, and ENCFF560BZE for alignment information of individual sequencing reads from H3K27me3 ChIP-seq experiments. Whole-genome Bisulfite Sequencing (WGBS) data are from ENCODE with accession number ENCFF847OWL, ENCFF390OZB, ENCFF064GJQ, and ENCFF369YQW. The active promoters used are selected based on HepG2 RNA-seq data from ENCODE under accession number ENCSR000CPC, ENCSR000CPE, and ENCSR000CPF. Source data are provided with this paper.

## Code availability
Source code for analysis is available at https://github.com/YinLabTJ/nanoHiMe.

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

## Acknowledgements

The authors are grateful to Drs. F. Zhu, L. Xiong and members of Yin laboratories for scientific suggestions and insightful comments during manuscript preparation. This work has been supported by the National Key Research and Development Program of China (2021YFC2701400 to Y. Y.), the National Natural Science Foundation of China (32070606 to Y. Y.), Shanghai Blue Cross Brain Hospital Co., Ltd. And Shanghai Tongji University Education Development Foundation, and the 2019 Thousand Youth Talents Plan of China to Y. Y.

## Author contributions

Y.Y. conceived the project. X.Y. carried out all the experiments with the help of Y.Y. and X.L. Z.X. established the pipeline for analyzing nanoHiMe-seq data with the help of Y.Y. M.L. and Z.X. analyzed CUT&Tag and CUT&Tag-BS data. K.W. designed the construct of proteinA-Hia5 for protein production. Y.Y. wrote the manuscript with the help of Z.X., X.Y., and M.L.

## Competing interests

The authors declare no competing interests.
