## [Peer Review File · Nature Communications]

REVIEWER COMMENTS

Reviewer #1 (Remarks to the Author):

Yue et al. present an interesting method for simultaneous detection of 6mA and 5mC and apply it for mapping histone modifications and DNA methylation.

The presented work is good and my comments are listed below:

- 1. How does nanoHiMe-seq compare with and differ from nanoNOMe (Lee et al. 2020) and DiMeLo-seq (Altemose et al. 2021)?**
- 2. Have you considered trying PEPPER-Margin-DeepVariant for phasing and allele-specific analyses?**
- 3. For phasing, how good were the results in phase block lengths and phasing quality?**
- 4. Is there a reason that the authors performed sheared 10 kb instead of going for longer?**
- 5. Is training emissions enough? Is it worth considering a signalAlign (Rand et al. 2017) or Taiyaki approach for training?**

My best wishes to the authors.

Reviewer #2 (Remarks to the Author):

The authors present a novel method, nanoHiMe-Seq, which aims to detect both DNA modifications and histone modifications simultaneously in the same DNA molecule. The ability to detect DNA modifications and specific histone modifications in the same long sequence read would represent a significant technical advance in the field, as would the reported ability to perform such analysis in an allele-specific manner as claimed. Thus the reviewer is broadly positive to the article and the results shown support that nanoHiMe-seq works and has potential benefits for the field, however, in its current form the authors have not convinced this reviewer of the novelty, robustness and broad applicability of nanoHiMe-Seq.

MAJOR COMMENTS

The paper is very poorly written and structured making a difficult and confusing read. As the authors present a technical advance far more about the actual method should be presented coherently at the beginning of the results section and not be scattered in the methods.

The figures throughout are so small as to be illegible in many cases.

The authors are disingenuous in their reference to competing technologies in the field. Although referencing ChIP-BS-Seq from 2012, they make no reference to far more recent and relevant advances including CUT-Tag-BS, nanoNOMe, and scNOMe-Seq. Even if nanoHiMe allows detection of specific histone modifications, it is with the aforementioned techniques that nanoHiMe-Seq is competing. Moreover, the advent of EM-Seq as an alternative to BS-Seq renders the criticism of ChIP-BS less relevant.

Consequently, this new technique must be benchmarked against other relevant techniques (at least one) in the same cell line to clarify its benefits in terms of accuracy, practicality and cost-effectiveness (several of the aforementioned far more straightforward than

nanoHiMe). A PRC2 knockout would also provide a biologically relevant control in addition to the IgG technical control, i.e. what is the false positive rate when little or none of the target histone modification exists. Finally, assessment of the accuracy of nanoHiMe with less starting material is critical to assessing its broader relevance beyond its use in cancer cell lines.

Why were HepG2 cells chosen and not the lymphoblastic cell lines used for most other nanopore technique development studies? This would have allowed simple direct comparison.

The ability to detect two different DNA modifications in the same read is not novel (CpG and GpC in NOME) as claimed.

The ability to phase and do allelic specific assignment of DNA methylation is not novel. Additionally, the utility of allele-specific epigenetic profiling would have been best shown on the X chromosome in female cells. This should be considered in any further experiments. HepG2 is male.

MINOR COMMENTS:

- Poorly written throughout including the abstract, needs proof-reading
- Enlarge all images (but especially gel images) in Supp Fig 1 to allow better assessment of digestion quality

Reviewer #3 (Remarks to the Author):

In this study, the authors describe nanoHiMe-seq, a method which can simultaneously map histone modification and CpG methylation using long nanopore reads. nanoHiMe-seq leverages adenine methyltransferase pA-Hia5 to introduce adenine methylation in the neighboring regions of antibody-targeted modified nucleosomes. Then, nanoHiMe-seq profiles the introduced 6mAs together with CpG methylation from the sequenced nanopore reads using an extended version of HMM. nanoHiMe-seq enables simultaneous analysis of haplotype-aware histone modifications and DNA methylation.

Simultaneously genome-wide mapping of histone modification and DNA methylation is a very relevant research question in epigenetics community. However, about this article, I do have several serious concerns. According to the concerns which I listed below, I do not recommend this manuscript to be published in Nat Commun.

Major

1. I notice that there are two previous studies, DiMeLo-seq [1] and BIND&MODIFY [2], which both are very similar to this article regarding the experimental design and the analysis, and both were preprinted in July. While this article/nanoHiMe-seq uses the adenine methyltransferase pA-Hia5, only experiments antibodies against histone modifications (H3K4me3 and H3K27me3), --DiMeLo-seq also leverages pA-Hia5. Besides histone modifications/variants, DiMeLo-seq was experimented to map lamina associated domains, and CTCF binding sites. In the DiMeLo-seq article, single-molecule DNA-protein interactions and endogenous CpG methylation were also jointly analyzed. --BIND&MODIFY uses a different methyltransferase pA-M.EcoGII. In the BIND&MODIFY article, antibody against H3K27me3 and Anti-CTCF antibody are used to map histone modifications and CTCF binding sites, respectively. For these reasons, I believe this article has very limited novelty. The authors should justify

their work in comparison to these two previous studies.

References

[1] Altemose N, Maslan A, Smith O K, et al. DiMeLo-seq: a long-read, single-molecule method for mapping protein-DNA interactions genome-wide[J]. bioRxiv, 2021.

[2] Weng Z, Ruan F, Chen W, et al. Long-range single-molecule mapping of chromatin modification in eukaryotes[J]. bioRxiv, 2021.

2. I believe one major work in this article is the implementation of an extended version of HMM which enables to predict 6mAs and 5mCpGs at the same time, while in the two previous studies, pre-trained models of ONT Guppy/Megalodon were used directly for 5mC and 6mA detection.

-- Regarding this, I think the authors basically followed the pipeline of nanopolish to train their new model. They expanded the emission lexicon, fit new gaussian mixture models, and re-implemented the call-methylation module using Perl/R. However, according to the description in the manuscript, it is still not clear enough how this new version of HMM differs from nanopolish. I suggest the authors emphasize the differences in Main text section and Methods section, since I believe this new version of HMM is a major contribution in their work.

-- The authors should also describe more about why they implement this new version of HMM. Does considering the co-influence of 6mA and 5mC signals in HMM significantly improve the performance in terms of accuracy and AUC? How about training a 6mA model (using Hia5- and Hia5+ PCR) and a 5mC model (using M.SssI- and M.SssI+ PCR) separately? Also, how this new HMM perform compared to nanopolish in 5mCpG calling? Does this new HMM outperform the pre-trained Guppy/Megalodon models in 5mC and 6mA calling?

-- The authors declare that nanoHiMe has the train-model module in README, however there is no interface to run the nanoHiMe train-model module in the provided source code.

3. It is not clear that how many coverage of nanopore reads the authors used in the analysis. Also, it would be better if the authors suggest a minimum requirement of nanopore reads to be sequenced to perform this genome-wide histone modification detection.

-- Page 4 line 82, 58% calls are kept when using 1 as threshold. In my view, it seems that too many calls are treated as ambiguous calls, which may increase the requirement of nanopore reads to be sequenced.

-- page 19 line 419 and page 20 line 436, when "coverage ≥ 20 " is used for further analysis, how many (portion of) sites and regions can be analyzed across the whole genome?

-- What is the mapping rate of the nanopore reads?

4. The manuscript in its present form in general is not well-organized and well-written. I strongly think that this manuscript should be re-organized and get English proofreading.

-- There are no subsections in the Main text section. There should be subsections for Introduction, Results, and Discussion at least.

-- The manuscript suffers from poor English. In particular, there are too many long sentences in it, which make the results and the methodology very difficult to understand. For example, page 3 line 51-57; page 5 line 105-108, 111-114, 115-116 "have been reported high selectivity for accessible DNA"; page 19 line 426 "the even alignment"; page 22 line 485-487; page 24 line 530-532, 540-543; page 25 line 549 "calling of methylation CpG sites", line 557-563.

Minor

- 1. Page 5 line 90, Which reference does "Isac et al.," cite?**
- 2. Fig. 2b, how is the "50%" in "across top 50% peaks" determined? Why not use all peaks?**
- 3. Page 22 line 500, what is "the same region from the E. coli K12 MG1655 genome" exactly?**

General response to reviewers

We greatly appreciate all referees' thoughtful comments and their very constructive advices. While all reviewers provided positive remarks, each referee also raised a number of concerns, which we have addressed through additional experiments and computational analysis and by rewriting the text. Main changes include:

- A systematic benchmarking of different computational tools, showing the high performance of the tool developed in this study
- Additional experiments to confirm the accuracy and high-sensitivity of nanoHiMe-seq
- More computational analysis showing nanoHiMe-seq is able to jointly profile chromatin features at low coverage depths and thus a low-cost method
- Phasing and allele-specific analyses using PEPPER-Margin-DeepVariant
- Extensively rewriting of the manuscript based on the formatting guidelines of Nature Communications and according to the suggestions from the referees

In the following pages is our point-by-point response to all specific comments of the reviewers. The reviewers' comments are in black, and our response to them is in blue. We sincerely thank the three referees for their great input that helped us make significant improvements in this revised manuscript.

N.B Since the reordering and restructuring of the manuscript was substantial, we have written bullet points of our major changes to the manuscript, rather than including a 'track changes' document. Line numbering refers to the revised manuscript.

Reviewer #1 (Remarks to the Author):

Yue et al. present an interesting method for simultaneous detection of 6mA and 5mC and apply it for mapping histone modifications and DNA methylation.

The presented work is good and my comments are listed below:

1. How does nanoHiMe-seq compare with and differ from nanoNOMe (Lee et al. 2020) and DiMeLo-seq (Altemose et al. 2021)?

We apologize for lacking comparison of nanoHiMe-seq to nanoNOMe and DiMeLo-seq in the original manuscript. We have now clarified in the text (**p.4, the first paragraph**) that nanoNOMe uses GpC methyltransferase M.CviPI to mark cytosines in open chromatin regions, and employs the computational tool – nanopolish to call the exogenously labeled cytosines as well as the endogenous methylated cytosines. In contrast, DiMeLo-seq and nanoHiMe-seq use N6-adenine methyltransferase Hia5 to label adenines proximal to chromatin proteins of interest. Different from DiMeLo-seq that employs ONT-provided computational tool - Megalodon to call the methylated adenines (6mAs) and methyl-cytosines (5mCs), nanoHiMe-seq uses a home-made computational tool - nanoHiMe, which simultaneously detects CpG and adenine methylation using a hidden Markov model.

As reported by previous study, the utility of nanoNOMe to investigate chromatin features is limited by factors including the sporadic occurrence and linear clustering of GpC dinucleotides across the human genome, and the endogenous cytosine methylation. Different from GpC, the devoid of endogenous methylation and the much higher occurrence frequency make adenine an ideal candidate to be marked in the chromatin regions of interest. So nanoHiMe-seq and DiMeLo-seq are more suitable to be used to investigate chromatin features that have different breadths, from ~5 nucleosomes for H3K4me2 to hundreds in H3K27me3 domains. This is now stated in the main text (**p.4, the first paragraph**).

Since the performance of Megalodon at simultaneously predicting 6mA and 5mC hasn't been systematically evaluated, we hence developed a computational tool - nanoHiMe and performed a systematic benchmarking of the two tools for 6mA and 5mC detection from individual nanopore sequencing reads. We found that, although Megalodon performed as well as nanoHiMe at identifying mCpG- and 6mA-containing sites from DNA with only one type of modification, nanoHiMe substantially outperformed Megalodon at predicting methylations from DNA with co-occurrence of 6mA and mCpG, and was more robust to explore chromatin features in realistic nanoHiMe-seq data. The results are now included in **Fig. 2c-d, 3a, 4b and 4d**, and described in **p.7 the first and the second paragraphs, p.8 the second paragraph, p.10 lines 252–264 and p.11 lines 282-287**.

2. Have you considered trying PEPPER-Margin-DeepVariant for phasing and allele-specific analyses?

We thank Reviewer #1's suggestion and have now performed phasing analysis using PEPPER-Margin-DeepVariant. We found that, although phase block lengths from WhatsHap and

PEPPER-Margin-DeepVariant were similar, the switch error rate and hamming error rate from PEPPER-Margin-DeepVariant were significantly lower than those from WhatsHap, thus resulted in a higher phasing quality. We have now re-analyzed the allele-specific patterns of DNA methylation, H3K4me3 and H3K27me3 based on the phasing information from PEPPER-Margin-DeepVariant, and updated the analysis results in **Fig. 6b-c** (Fig.3c and 3d in previous version) and Supplementary **Fig.7-9**.

3. For phasing, how good were the results in phase block lengths and phasing quality?

The average phase block length N50, switch error rate and hamming error rate from PEPPER-Margin-DeepVariant are 306 kb, 0.97% and 5.01%, respectively. We have now included the statistics from each chromosome in Supplementary **Table 4** and described in **p.28 the first paragraph**.

4. Is there a reason that the authors performed sheared 10 kb instead of going for longer?

We observed that the yield from one flowcell was usually negatively correlated with the DNA length sequenced. As the super long reads were not desperately required for this study, we tried to get more data from each flowcell by shearing DNA to 10 kb.

5. Is training emissions enough? Is it worth considering a signalAlign (Rand et al. 2017) or Taiyaki approach for training?

We thank Reviewer #1's suggestion. To assess the performance of our training step, we first compared our learned parameters from a DNA sample lacking methylation or treated with M.SssI to corresponding parameters of individual *k*-mers provided by nanopolish, a widely used computational tool that applies an HMM to call 5mC in CpG and GpC contexts. We found that the parameters learned by nanoHiMe were highly consistent with the those provided by nanopolish. Additionally, we trained the parameters of emission distribution for individual *k*-mers using signalAlign in DNA samples treated with M.SssI, Hia5, or both, which were previously used by nanoHiMe for parameter learning. We found that the parameters learned using nanoHiMe also correlated well with those learned from signalAlign for each DNA sample. We have now included the results in Supplementary **Fig. 3a-b** and described in **p.6 lines 133-141**.

My best wishes to the authors.

We thank Reviewer #1 for the encouragement.

Reviewer #2 (Remarks to the Author):

The authors present a novel method, nanoHiMe-Seq, which aims to detect both DNA modifications and histone modifications simultaneously in the same DNA molecule. The ability to detect DNA modifications and specific histone modifications in the same long sequence read would represent a significant technical advance in the field, as would the reported ability to perform such analysis in an allele-specific manner as claimed. Thus the reviewer is broadly positive to the article and the results shown support that nanoHiMe-seq works and has potential benefits for the field, however, in its current form the authors have not convinced this reviewer of the novelty, robustness and broad applicability of nanoHiMe-Seq.

We thank reviewer #2's generally positive comments on our manuscript and fully agree that the ability to detect DNA methylation and specific histone modifications in the same long sequence read represents a significant technical advance in the field of epigenomics. To address the reviewer's concern, we compared the performance of our computational tool to the existing tools (**Fig. 2c-d, 3a, 4b, 4d**), performed a systematic benchmarking of nanoHiMe-seq and a newly developed method - CUT&Tag-BS for joint profiling of histone modification and DNA methylation (**Fig. 4c-d**; Supplementary **Fig. 6d-f**), and assessed the minimum sequencing depth required for nanoHiMe-seq (**Fig. 5**). We demonstrated the ease, robustness, sensitivity and cost-effectiveness of nanoHiMe-seq in the revised manuscript. We anticipate that nanoHiMe-seq will be widely used to study epigenetics in diverse areas of biological research.

MAJOR COMMENTS:

The paper is very poorly written and structured making a difficult and confusing read. As the authors present a technical advance far more about the actual method should be presented coherently at the beginning of the results section and not be scattered in the methods. The figures throughout are so small as to be illegible in many cases.

We agree that the original version was too short and unclear. We've now restructured our manuscript based on the formatting guidelines of Nature Communications, and extensively rewritten the text with clarity in mind. We've also asked two colleagues who were not familiar with the work to read the text and comment especially on readability. Moreover, as suggested by the reviewer, we have now coherently introduced and assessed our methods at the beginning of results section. We believe that the manuscript reads now better, and is also more accessible to the general reader.

We apologize for the low-resolution figures and have now replaced all of them with high-quality ones.

The authors are disingenuous in their reference to competing technologies in the field. Although referencing ChIP-BS-Seq from 2012, they make no reference to far more recent and relevant advances including CUT-Tag-BS, nanoNOME, and scNOME-Seq. Even if nanoHiMe allows detection of specific histone modifications, it is with the aforementioned techniques that nanoHiMe-Seq is competing. Moreover, the advent of EM-Seq as an alternative to BS-Seq renders the criticism of ChIP-BS less relevant.

We apologize for not mentioning the recent and relevant advances in epigenomic technologies in the original manuscript and have now added the references in the revised manuscript. We agree that nanoHiMe-seq is competing with the recent techniques including CUT&Tag-BS, nanoNOME and scNOME-seq, all of which have been developed to jointly analyze chromatin features in the same DNA molecules in a single assay. Although these advances are remarkable and have fueled the growth of the field of epigenomics, they are hampered by few problems. For instance, CUT&Tag-BS and scNOME-Seq have the disadvantages of DNA damages, complexity reduction and biases introduced by the bisulfite treatment, even under mild conditions; Enzymatic Methyl-seq (EM-seq) overcomes the limitation of bisulfite treatment, but the difficulties to prepare hyperactive enzyme(s) curb its broad application.

nanoNOME is a nanopore sequencing based method that uses GpC methyltransferase M.CviPI to mark cytosines in open chromatin regions. The utility of nanoNOME to investigate chromatin features is limited by factors including the sporadic occurrence and linear clustering of GpC dinucleotides across the human genome and the endogenous cytosine methylation. Due to its lack of endogenous methylation and its much higher frequency, at almost one in every two DNA base-pairs, adenine is an ideal candidate for marking regions of interest across the human genome. So nanoHiMe-seq, which employs N6-adenine methyltransferase to label adenines proximal to chromatin proteins of interest, is in principle more suitable than nanoNOME to be applied to jointly investigate chromatin features from single DNA molecules.

This is now stated in the main text (p.3 lines 47–62; p.4 lines 69–77).

Consequently, this new technique must be benchmarked against other relevant techniques (at least one) in the same cell line to clarify its benefits in terms of accuracy, practicality and cost-effectiveness (several of the aforementioned far more straightforward than nanoHiMe). A PRC2 knockout would also provide a biologically relevant control in addition to the IgG technical control, i.e. what is the false positive rate when little or none of the target histone modification exists. Finally, assessment of the accuracy of nanoHiMe with less starting material is critical to assessing its broader relevance beyond its use in cancer cell lines.

We thank Reviewer #2's suggestion and have now performed a systematic benchmarking of nanoHiMe-seq and CUT&Tag-BS for CpG methylation and H3K27me3 detection. We found that patterns of 6mA signals from H3K27me3 nanoHiMe-seq echoed the profiles generated by CUT&Tag-BS for H3K27me3, and that the methylation levels measured by nanoHiMe-seq and CUT&Tag-BS were also highly consistent. In addition, when comparing CUT&Tag-BS with CUT&Tag, we found that some of H3K27me3 peaks identified by CUT&Tag were not detected in CUT&Tag-BS, probably due to the DNA degradation resulting from bisulfite treatment. When comparing nanoHiMe-seq with CUT&Tag, we found that the enrichment of 6mA signals was detected in the peak regions identified by CUT&Tag-BS and in the regions with weak peaks identified only by CUT&Tag, thus highlighting the high sensitivity of nanoHiMe-seq. Moreover, unlike CUT&Tag-BS that only measures the CpG methylation levels at the peak regions, nanoHiMe-seq also measures the methylation levels of CpG sites outside of the peak regions, enabling the investigation of the relationship between histone modifications and methylation of

the CpGs inside and outside the peak regions. The results are now included in **Fig. 4c-d** and Supplementary **Fig. 6b-f**, and discussed in **lines 271–284** and **lines 380–397**.

Like CUT&Tag, the workflow of nanoHiMe-seq is quite simple, and the procedure is easily implemented in a standardized approach. The entire procedure from cell harvest to library preparation can be performed in one day, and profiling of histone modification and CpG methylation could be obtained using approximately 10× coverage depth, corresponding to the reads produced by only one or two MinION flowcells from 100,000 – 200,000 cells. The starting cell number could even decrease if lower amounts of gDNA loaded to each MinION flowcell is appreciated in the near future. This now is stated in **p.11 the second paragraph** and **p.14 the first paragraph**. The results are included in **Fig. 5**.

As suggested by the reviewer, we've performed H3K27me3 nanoHiMe-seq in GM12878 cells treated with EZH2 inhibitor EPZ6438, in which H3K27 methylation was severely depleted. After 7 days treatment, we found the H3K27me3 was almost undetectable by western blot, and the 6mA signals from these cells were very sparse, with a 6mA-containing site being detected in approximately 16,400 bp along the genome under LLR cutoff of 32. The results are included in **Fig. 3d** and Supplementary **Fig. 4e-f** and described in **lines 218–222**.

Why were HepG2 cells chosen and not the lymphoblastic cell lines used for most other nanopore technique development studies? This would have allowed simple direct comparison.

The ability to detect two different DNA modifications in the same read is not novel (CpG and GpC in NOME) as claimed.

The ability to phase and do allelic specific assignment of DNA methylation is not novel. Additionally, the utility of allele-specific epigenetic profiling would have been best shown on the X chromosome in female cells. This should be considered in any further experiments. HepG2 is male.

We agree and have now moderated these statements throughout. We've also conducted nanoHiMe-seq in GM12878 cells, a female lymphoblastic cell line, and performed allele-specific analysis of H3K27me3 and CpG methylation across the genome. The results are now included in Supplementary **Fig. 7** and Supplementary **Table 5–6** and described in **lines 327–330** and **345–349**.

MINOR COMMENTS:

- Poorly written throughout including the abstract, needs proof-reading

We have rewritten the text and got English proofreading.

- Enlarge all images (but especially gel images) in Supp Fig 1 to allow better assessment of digestion quality

We've now replaced Supp Fig 1 with a high-quality figure.

Reviewer #3 (Remarks to the Author):

In this study, the authors describe nanoHiMe-seq, a method which can simultaneously map histone modification and CpG methylation using long nanopore reads. nanoHiMe-seq leverages adenine methyltransferase pA-Hia5 to introduce adenine methylation in the neighboring regions of antibody-targeted modified nucleosomes. Then, nanoHiMe-seq profiles the introduced 6mAs together with CpG methylation from the sequenced nanopore reads using an extended version of HMM. nanoHiMe-seq enables simultaneous analysis of haplotype-aware histone modifications and DNA methylation.

Simultaneously genome-wide mapping of histone modification and DNA methylation is a very relevant research question in epigenetics community. However, about this article, I do have several serious concerns. According to the concerns which I listed below, I do not recommend this manuscript to be published in Nat Commun.

We thank reviewer #3 for his/her agreement with that simultaneous mapping of histone modification and DNA methylation is a very relevant research question in epigenetics community. In the revised manuscript, we have tried our best to address the reviewer's concern by additional computational analysis and a systematic benchmarking of relevant techniques for histone modification and CpG methylation detection. We thoroughly compared the performance of our computational tool and Megalodon, a tool that was provided by Oxford Nanopore Technologies (ONT) and used by DiMeLo-seq and BIND&MODIFY for profiling the genomic localization of the proteins of interest and the methylation status of the underlying DNA. We found that our computational tool substantially outperformed Megalodon, making nanoHiMe-seq more robust and sensitive than DiMeLo-seq and BIND&MODIFY to jointly profile chromatin features from the same DNA molecules in a single assay.

Major:

1. I notice that there are two previous studies, DiMeLo-seq [1] and BIND&MODIFY [2], which both are very similar to this article regarding the experimental design and the analysis, and both were preprinted in July. While this article/nanoHiMe-seq uses the adenine methyltransferase pA-Hia5, only experiments antibodies against histone modifications (H3K4me3 and H3K27me3),

--DiMeLo-seq also leverages pA-Hia5. Besides histone modifications/variants, DiMeLo-seq was experimented to map lamina associated domains, and CTCF binding sites. In the DiMeLo-seq article, single-molecule DNA-protein interactions and endogenous CpG methylation were also jointly analyzed.

--BIND&MODIFY uses a different methyltransferase pA-M.EcoGII. In the BIND&MODIFY article, antibody against H3K27me3 and Anti-CTCF antibody are used to map histone modifications and CTCF binding sites, respectively.

For these reasons, I believe this article has very limited novelty. The authors should justify their work in comparison to these two previous studies.

References

[1] Altomose N, Maslan A, Smith O K, et al. DiMeLo-seq: a long-read, single-molecule method for mapping protein-DNA interactions genome-wide[J]. bioRxiv, 2021.

[2] Weng Z, Ruan F, Chen W, et al. Long-range single-molecule mapping of chromatin modification in eukaryotes[J]. bioRxiv, 2021.

We thank Reviewer #3's suggestion and have now cited the two references. Comparing to the two studies, the main additional contributions of this article include 1) the development of a computational tool to jointly detect 5mC and 6mA from single DNA molecule with a hidden Markov model (**Fig. 1**; Supplementary **Fig. 2**; **Methods** under heading "Model training"; **lines 115–132**), 2) systematic evaluation of the performance of different computational tools at identifying mCpG and/or 6mA (**Fig. 2c-d, 3a, 4b and 4d**), 3) a benchmarking of nanoHiMe-seq and a newly developed technique - CUT&Tag-BS for simultaneous profiling of CpG methylation and histone modifications (**Fig. 4c-d**; Supplementary **6b-f**), and 4) assessment of the minimum coverage depth required for nanoHiMe-seq (**Fig. 5**).

We demonstrated the ease, robustness, high-sensitivity and cost-effectiveness of nanoHiMe-seq in the revised manuscript, and anticipate that nanoHiMe-seq will be widely used to study epigenetics in diverse areas of biological research. We also highlighted the strengths and weakness of state-of-the-art methods to jointly profile chromatin features using nanopore or Illumina sequencing technologies, which will benefit the epigenetics community.

2. I believe one major work in this article is the implementation of an extended version of HMM which enables to predict 6mAs and 5mCpGs at the same time, while in the two previous studies, pre-trained models of ONT Guppy/Megalodon were used directly for 5mC and 6mA detection.

-- Regarding this, I think the authors basically followed the pipeline of nanopolish to train their new model. They expanded the emission lexicon, fit new gaussian mixture models, and re-implemented the call-methylation module using Perl/R. However, according to the description in the manuscript, it is still not clear enough how this new version of HMM differs from nanopolish. I suggest the authors emphasize the differences in Main text section and Methods section, since I believe this new version of HMM is a major contribution in their work.

We thank Reviewer #3's suggestion, and have now stated the differences between nanoHiMe and nanopolish in main text (**lines 123–132**). Nanopolish learned the parameters of emission distribution for individual k -mers from fully methylated DNA and thus was unable to train the k -mers with a mixture of methylation patterns (TMGACG, M indicates methylated cytosine). So, when testing for methylation, it was assumed that all sites in the group have the same methylation status. NanoHiMe also followed the same principal for calling CpG methylation as it was expected that the methylation status of nearby CpG sites was correlated. However, in nanoHiMe-seq data, it is unlikely that the adenines in a site proximal to modified nucleosomes were all methylated or unmethylated. In order to call adenine methylation from all possible contexts, nanoHiMe learned the parameters from DNA templates with partial adenine methylation, which enabled us to obtain parameters of k -mers with 6mA in all possible contexts. The partial adenine methylation made it impossible to precisely assign the learned parameters to k -mers that were derived from a four-letter k -mer, but contain methyladenine(s) in different positions, such as TZCACG, TACZCG and TZCZCG, where Z denotes 6mA. To overcome this

limitation, we grouped such k -mers as a new k -mer \bar{k} and assigned the parameters from them to \bar{k} . As a result, nanoHiMe gained the ability to accurately identify the sites with fully methylated adenines, a mixture of methylation patterns, or free of methylation as a 6mA-containing site or a non-6mA site, but lost the ability to predict adenine methylation at base resolution.

-- The authors should also describe more about why they implement this new version of HMM. Does considering the co-influence of 6mA and 5mC signals in HMM significantly improve the performance in terms of accuracy and AUC? How about training a 6mA model (using Hia5- and Hia5+ PCR) and a 5mC model (using M.SssI- and M.SssI+ PCR) separately?

As described above, we have now clarified in the main text (**lines 123–132**) that we used the new version of HMM to correctly identify the sites with fully methylated adenines, a mixture of methylation patterns, or free of methylation as a 6mA-containing site or non-6mA site.

We have now compared the performance of nanoHiMe at predicting methylations by considering or not considering the co-influence of 6mA and 5mC, and found that it performed slightly better at identifying 6mA- and mCpG-containing sites from DNA with two types of methylations by considering the co-influence (6mA at 50-bp sites AUC: 0.995 vs 0.9799; mCG AUC: 0.9222 vs 0.9221). This is now stated in **lines 182–186**.

Also, how this new HMM perform compared to nanopolish in 5mCpG calling? Does this new HMM outperform the pre-trained Guppy/Megalodon models in 5mC and 6mA calling?

We thank reviewer #3's suggestion and have now performed a systematic benchmarking of nanoHiMe, nanopolish and Megalodon for mCpG and/or 6mA detection. We found that Megalodon performed slightly better than nanopolish and nanoHiMe at predicting CpG methylation on DNA that didn't contain 6mAs. However, when calling CpG methylation from DNA that contained 6mAs, nanoHiMe and nanopolish substantially outperformed Megalodon, with nanoHiMe showing the best performance (**Fig. 2c**; described in **lines 148–169**). We further compared the performance of nanoHiMe, nanopolish and Megalodon at calling CpG methylation in nanopore reads obtained from nanoHiMe-seq experiments, and found methylation levels calculated based on the calls from nanoHiMe correlated best with WGBS measurements (**Fig. 4b**; described in **lines 239–259**). Moreover, we found that methylation levels predicted by nanoHiMe were aligned better than those from Megalodon with measurements from CUT&Tag-BS for the CpGs in H3K27me3-marked regions (**Fig. 4d**; described in **lines 272–282**).

When comparing the performance of nanoHiMe and Megalodon at identifying 6mA-containing sites, we found that the performance of both nanoHiMe and Megalodon was highly correlated with the length of the sites assessed, with better performance in predicting adenine methylation for longer sites. When evaluating sites that did not harbor mCpG(s) and were no shorter than 40 bp, nanoHiMe performed slightly better than Megalodon. When the evaluated sites also contained mCpG(s), nanoHiMe substantially outperformed Megalodon (**Fig. 2d**; described in **lines 170–182**). Moreover, we found that nanoHiMe was more robust than Megalodon to identify the high-confidence 6mA-containing sites in nanoHiMe-seq data (**Fig. 3a**; described in **lines 194–206**).

-- The authors declare that nanoHiMe has the train-model module in README, however there is no interface to run the nanoHiMe train-model module in the provided source code.

We apologize for missing the command lines used for model training. We have now added a step-by-step instruction for training the parameters of individual k -mers in README.

3. It is not clear that how many coverage of nanopore reads the authors used in the analysis. Also, it would be better if the authors suggest a minimum requirement of nanopore reads to be sequenced to perform this genome-wide histone modification detection.

We have now clarified that the average coverage depth of H3K27me3 nanoHiMe-seq was 56× in HepG2 cells and 47× in GM12878 cells, and the average converge depth of H3K4me3 nanoHiMe-seq was 40× (Supplementary **Table 2**; described in **Methods section** under heading “**Nanopore sequencing data preprocessing**”). We have also down-sampled nanopore reads from nanoHiMe-seq experiments to see the minimum coverage depth required by nanoHiMe-seq for joint profiling of DNA methylation and histone modification. We found that the patterns of 6mA signals from H3K27me3 nanoHiMe-seq resembled the profiles generated by CUT&Tag and ChIP-seq at 10× coverage, corresponding to the reads produced by only one or two MinION flowcells from about 100, 000 – 200, 000 cells. At the 10× coverage depth, we found the correlation coefficient between CpG methylation measurements from nanoHiMe and WGBS was as high as 0.82. The results are now included in **Fig. 5**, and described under heading “**nanoHiMe-seq sensitively profiles chromatin features at low coverage depths**”.

-- Page 4 line 82, 58% calls are kept when using 1 as threshold. In my view, it seems that too many calls are treated as ambiguous calls, which may increase the requirement of nanopore reads to be sequenced.

We thank reviewer #3 for pointing out this important issue. Previously, nanoHiMe called adenine methylation in a 11-bp window, and could identify 58% 6mA-containing sites using a cut-off of 1. When revising the manuscript, we found that the performance of both nanoHiMe and Megalodon was highly correlated with the length of the sites assessed, with better performance in predicting adenine methylation for longer sites (**Fig. 2d**; described in **lines 170–182**). In the revised manuscript, we extended the 11-bp window to 50-bp for calling adenine methylation, which resulted in correctly identifying 86.6% 6mA-containing sites using a cut-off of 8. The contingency table is now included in Supplementary **Table 3**.

-- page 19 line 419 and page 20 line 436, when “coverage ≥ 20 ” is used for further analysis, how many (portion of) sites and regions can be analyzed across the whole genome?

-- What is the mapping rate of the nanopore reads?

We have now clarified the breadth and depth of nanoHiMe-seq coverage for H3K27me3 and H3K4me3 in Supplementary **Fig. 4c**. We found that 90.63% and 90.2% of the genome were covered by a least 20 nanopore reads in H3K27me3 nanoHiMe-seq experiment from HepG2 cells and GM12878 cells, respectively. 88.93% of the genome was covered by a least 20 reads in H3K4me3 nanoHiMe-seq experiment.

The average mapping rate of nanopore reads from nanoHiMe-seq was 83.7%. The average mapping rate of nanopore reads from PCR⁺ M.SssI⁻ Hia5⁻, PCR⁺ M.SssI⁺ Hia5⁻, PCR⁺ M.SssI⁻ Hia5⁺, PCR⁺ M.SssI⁺ Hia5⁺ samples were 88.3%, 94.2%, 98%, and 87.5%, respectively. The statistics are now included in Supplementary **Table 2**.

4. The manuscript in its present form in general is not well-organized and well-written. I strongly think that this manuscript should be re-organized and get English proofreading.

-- There are no subsections in the Main text section. There should be subsections for Introduction, Results, and Discussion at least.

-- The manuscript suffers from poor English. In particular, there are too many long sentences in it, which make the results and the methodology very difficult to understand. For example, page 3 line 51-57; page 5 line 105-108, 111-114, 115-116 “have been reported high selectivity for accessible DNA”; page 19 line 426 “the even alignment”; page 22 line 485-487; page 24 line 530-532, 540-543; page 25 line 549 “calling of methylation CpG sites”, line 557-563.

We apologize for the obscurity and ambiguity of the original manuscript. As described above in response to ref #2, we have now restructured our manuscript based on the formatting guidelines of Nature Communications and extensively rewritten the text with clarity in mind. Moreover, we also asked two colleagues who were not familiar with the work to read the text and comment especially on readability. We believe that the manuscript reads now better, and is also more accessible to the general reader.

Minor:

1. Page 5 line 90, Which reference does “Isac et al.,” cite?

We have now cited the reference (**line 246**).

2. Fig. 2b, how is the “50%” in “across top 50% peaks” determined? Why not use all peaks?

We have now clarified that almost all of the top 50% peaks in ChIP-seq overlapped with those identified in CUT&Tag experiments and *vice versa*, so these peaks represent high-confident H3K27me3-marked regions. The remaining peaks, especially the ones in the fourth quartile, were often observed only in either ChIP-seq or CUT&Tag, representing weak sites or noises (**Methods section** under heading “**Comparison between nanoHiMe-seq, CUT&Tag and ChIP-seq or between nanoHiMe-seq and WGBS**”).

3. Page 22 line 500, what is “the same region from the E. coli K12 MG1655 genome” exactly?

We have now clarified that we used the region from 50, 000 bp to 3, 250, 000 bp of the *E. coli* K12 MG1655 genome for the parameters training (**line 818**).

REVIEWERS' COMMENTS

Reviewer #1 (Remarks to the Author):

Thank you for addressing my comments. I don't have any further questions.

Reviewer #2 (Remarks to the Author):

The authors present a much revised version of the manuscript including new analyses and experiments to address my earlier concerns. These include new CUT&TAG-BS-seq experiments in HepG2 cells facilitating comparison with nanoHiMe-seq, nanoHiMe-seq in GM12878 cells and pharmacological knock-out of PRC2 activity.

These are substantial efforts that adequately address my concerns and support the authors claims of nanoHiMe-seq robustness and specificity. I also appreciate the changes to the text which place nanoHiMe-seq within the context of a rapidly expanding plethora of methods aiming for simultaneous detection of 5mC and histone marks.

I have no further comments for the authors

Reviewer #3 (Remarks to the Author):

First, I want to thank the authors' hard work on this revision. But, after reading the revised manuscript, and the authors' responses to me and other reviewers, I don't think this work can be published in Nature Communications.

1. The biological design of nanoHiMe-seq has no novelty compared to DiMeLo-seq, as both nanoHiMe-seq and DiMeLo-seq uses pA-Hia5 methyltransferase. Moreover, DiMeLo-seq has been evaluated on mapping lamina associated domains, CTCF binding sites and histone modifications/variants, while nanoHiMe-seq only focuses on histone modifications.

-- Also, because of this, the authors may need to change the title of the manuscript, re-write the main text to emphasize their main contribution.

2. The computational part of nanoHiMe-seq has very limited novelty. First, re-training an HMM model based on nanopolish to detect two modifications at the same time is not novel (nanoNOME, Lee et al. 2020). Second, although it seems that the authors have done tremendous work to evaluate nanoHiMe-seq on 5mC/6mA detection in comparison to nanopolish/Megalodon, nanoHiMe-seq didn't significantly improve the performances in most cases (according to section nanoHiMe-seq performance).

Reviewer #1 (Remarks to the Author):

Thank you for addressing my comments. I don't have any further questions.

Reviewer #2 (Remarks to the Author):

The authors present a much revised version of the manuscript including new analyses and experiments to address my earlier concerns. These include new CUT&TAG-BS-seq experiments in HepG2 cells facilitating comparison with nanoHiMe-seq, nanoHiMe-seq in GM12878 cells and pharmacological knock-out of PRC2 activity.

These are substantial efforts that adequately address my concerns and support the authors claims of nanoHiMe-seq robustness and specificity. I also appreciate the changes to the text which place nanoHiMe-seq within the context of a rapidly expanding plethora of methods aiming for simultaneous detection of 5mC and histone marks.

I have no further comments for the authors

Reviewer #3 (Remarks to the Author):

First, I want to thank the authors' hard work on this revision. But, after reading the revised manuscript, and the authors' responses to me and other reviewers, I don't think this work can be published in Nature Communications.

1. The biological design of nanoHiMe-seq has no novelty compared to DiMeLo-seq, as both nanoHiMe-seq and DiMeLo-seq uses pA-Hia5 methyltransferase. Moreover, DiMeLo-seq has been evaluated on mapping lamina associated domains, CTCF binding sites and histone modifications/variants, while nanoHiMe-seq only focuses on histone modifications.

-- Also, because of this, the authors may need to change the title of the manuscript, re-write the main text to emphasize their main contribution.

We thank Reviewer #3 again for his/her comments and consideration. It is worth noting that DiMeLo-seq was not published until our work was submitted and under peer review, and was in fact only submitted to Nature Methods 2 months prior. During the time, our work was protected by Scooping Protection policy of Nature Communications.

Although both DiMeLo-seq and nanoHiMe-seq use the protein A-N6-adenine methyltransferase fusion protein pA-Hia5 to label adenines proximal to chromatin proteins of interest, DiMeLo-seq uses Oxford Nanopore Technologies (ONT) provided computational tool - Megalodon to jointly detect the labelled adenines (6mAs) and the endogenous methylated CpGs (mCpGs) on individual nanopore reads, and nanoHiMe-seq uses the computational tool - nanoHiMe developed in this study to concurrently identify 6mAs and mCpGs on each read. When comparing the performance of Megalodon and nanoHiMe at detecting 6mAs and mCpGs from single nanopore reads, we observed that, nanoHiMe substantially outperformed Megalodon at calling 6mAs and mCpGs on the DNA that contains both 6mAs and mCpGs (CpG methylation: nanoHiMe AUC 0.922 vs Megalodon AUC 0.603; adenine methylation: nanoHiMe

AUC 0.992 vs Megalodon AUC 0.967 at 40bp; **Figure 1d-e**), although both tools performed quite well at identifying 6mAs and mCpGs on the DNA that only contains either 6mAs or mCpGs. Moreover, when comparing the performance of nanoHiMe and Megalodon at calling 6mAs and mCpGs in the realistic nanoHiMe-seq data, we found that methylation levels predicted by nanoHiMe were aligned significantly better than those from Megalodon with measurements from CUT&Tag-BS for the CpGs in H3K27me3-marked regions (Pearson's correlation coefficient r 0.871 vs 0.77; **Figure 3d**), and that nanoHiMe was more robust than Megalodon to identify the high-confidence 6mA-containing sites (The efficiency-to-noise ratio: nanoHiMe 10.85 vs Megalodon 6.85 at discovery efficiency of 0.05; **Fig. 2a**). Additionally, we also demonstrated the robustness, high-sensitivity and cost-effectiveness of nanoHiMe-seq in the manuscript, and anticipate that nanoHiMe-seq will be widely used to study functional coordination of epigenetic marks in various biological contexts.

As Reviewer #3 said, nanoHiMe-seq focuses on histone modifications and DNA methylation. So, we believe the title of our manuscript and the main text, both of which emphasize the utility of nanoHiMe-seq on joint profiling of histone modifications and DNA methylation, are well supported by our results, and appropriately describe the advances of our methods and the conclusions in current work. We don't think it is necessary to change them.

2. The computational part of nanoHiMe-seq has very limited novelty. First, re-training an HMM model based on nanopolish to detect two modifications at the same time is not novel (nanoNOMe, Lee et al. 2020). Second, although it seems that the authors have done tremendous work to evaluate nanoHiMe-seq on 5mC/6mA detection in comparison to nanopolish/Megalodon, nanoHiMe-seq didn't significantly improve the performances in most cases (according to section nanoHiMe-seq performance).

We apologize for the misunderstanding. First, the HMM model in this study was trained using our new pipeline, but not nanopolish, which could learn the parameters of k -mers from completely methylated DNA and was unable to train k -mers with mixture of methylation patterns, such as TMGACG with M denoting methylated cytosine. As a result, when calling CpG/GpC methylation by nanopolish, it was assumed that all sites in the group have the same methylation status. However, in nanoHiMe-seq data, it is unlikely that the adenines in a site proximal to modified nucleosomes were all methylated or unmethylated. In order to call adenine methylation from all possible contexts, nanoHiMe learned the parameters from DNA templates with partial adenine methylation, which enabled us to obtain parameters of k -mers with 6mA in all possible contexts, such as TZCACG, TACZCG and TZCZCG, where Z denotes 6mA. This has now been stated in main text (**p. 6, first paragraph**). Second, as described above, we do observe nanoHiMe performs better than Megalodon at calling 6mAs and mCpGs on the DNA that contain both 6mAs and mCpGs in testing data and realistic nanoHiMe-seq data. The results have been included in **Figure 1d-e**, **Fig. 2a** and **Figure 3d**.